# Computational models reveal how chloride dynamics determine the optimal distribution of inhibitory synapses to minimise dendritic excitability

**Christopher Brian Currin**[ID]*, **Joseph Valentino Raimondo**[ID]

Division of Cell Biology, Department of Human Biology, Neuroscience Institute and Institute of Infectious Disease and Molecular Medicine, Faculty of Health Sciences, University of Cape Town, Cape Town, South Africa

* chris.currin@gmail.com

**Data Availability Statement:** https://github.com/ChrisCurrin/chloride-dynamics-and-dendrites.

**Funding:** CBC is supported by the German Deutscher Akademischer Austauschdienst (DAAD,

## Abstract

Many neurons in the mammalian central nervous system have complex dendritic arborisations and active dendritic conductances that enable these cells to perform sophisticated computations. How dendritically targeted inhibition affects local dendritic excitability is not fully understood. Here we use computational models of branched dendrites to investigate where GABAergic synapses should be placed to minimise dendritic excitability over time. To do so, we formulate a metric we term the "Inhibitory Level" (IL), which quantifies the effectiveness of synaptic inhibition for reducing the depolarising effect of nearby excitatory input. GABAergic synaptic inhibition is dependent on the reversal potential for GABA$_A$ receptors (EGABA), which is primarily set by the transmembrane chloride ion (Cl$^-$) concentration gradient. We, therefore, investigated how variable EGABA and dynamic chloride affects dendritic inhibition. We found that the inhibitory effectiveness of dendritic GABAergic synapses combines at an encircled branch junction. The extent of this inhibitory accumulation is dependent on the number of branches and location of synapses but is independent of EGABA. This inhibitory accumulation occurs even for very distally placed inhibitory synapses when they are hyperpolarising–but not when they are shunting. When accounting for Cl$^-$ fluxes and dynamics in Cl$^-$ concentration, we observed that Cl$^-$ loading is detrimental to inhibitory effectiveness. This enabled us to determine the most inhibitory distribution of GABAergic synapses which is close to–but not at–a shared branch junction. This distribution balances a trade-off between a stronger combined *inhibitory influence* when synapses closely encircle a branch junction with the deleterious effects of increased Cl$^-$ by loading that occurs when inhibitory synapses are co-located.

## Author summary

Dendritic branches allow for a rich repertoire of computational capabilities for neurons within the brain. Inhibitory synaptic inputs, which utilise the neurotransmitter GABA,

https://www.daad.de/), the South African (SA) National Research Foundation (NRF, https://www.nrf.ac.za), and the University of Cape Town (UCT, https://www.uct.ac.za). JVR is supported the National Research Foundation of South Africa, a Wellcome Trust Seed Award (214042/Z/18/Z), the South African Medical Research Council and by the FLAIR Fellowship Programme (FLR\R1\190829): a partnership between the African Academy of Sciences and the Royal Society funded by the UK Government's Global Challenges Research Fund. The funders had no role in study design, data collection and analysis, decision to publish, or preparation of the manuscript.

**Competing interests:** The authors have declared that no competing interests exist.

refine and enhance dendritic computations. They are traditionally viewed with regards to their inhibitory effect on action potential generation at the neuronal cell body. Here, we studied the local effects of inhibitory synapses on excitability in dendrites. We also considered the dynamic nature of inhibition that deteriorates the longer it is active due to intracellular chloride ion loading. The central goal of our investigation was to find the best locations for multiple inhibitory synapses to maximise their combined inhibitory effectiveness on nearby excitation in the dendritic tree. We found that the optimal distribution is when inhibitory synapses closely encircle a branch junction, without being co-located at the junction itself. This maximises how their inhibitory influence combines whilst minimising the deleterious effects of chloride loading.

## Introduction

A typical layer 2/3 cortical pyramidal neuron can have thousands of synaptic sites from hundreds to thousands of neurons [1–3]. A neuron's ability to integrate synaptic input over multiple time scales and physical spatial domains is possible due to the physical structure and electrochemical properties of its dendrites [4–6]. Dendrites enable neurons to perform a broad array of possible computations [7]. Experimental and theoretical results over the last few decades have shifted the characterisation of dendrites from simply the input domain of neurons to a neuronal domain with rich computational complexity [8–13].

The theoretical understanding of signal propagation in dendrites was established by several seminal studies [14–19]. This understanding was grounded in the assumption that dendrites are passive. We now know that they also possess active properties driven by NMDA, voltage-gated calcium ($Ca^{2+}$), and voltage-gated sodium ($Na^+$) channels [20], which adds to the complexity of dendritic processing [6,21]. The traditional view of inhibition in dendrites has focused on its effects at the soma. That is, how does dendritic inhibition change the excitatory current that reaches the soma, which affects the activation of non-linear processes there or at the axon initial segment. Due to the experimental challenges involved in recording from dendrites, the somato-centric view has been more prevalent to-date. In contrast to this somato-centric viewpoint, it has now become clear that many neurons possess dendrites with non-linear conductances (e.g. NMDA receptors, voltage-gated $Ca^{2+}$, $Na^+$ and $K^+$ channels as well as HCN channels). It is now thought that dendritic branches could act as individual non-linear integration zones; meaning that single neurons could represent the biological equivalent of multi-level artificial neuronal networks [13,22,23]. This points to the importance of also considering a dendro-centric viewpoint, i.e. how does dendritic inhibition control local dendritic excitability?

In this vein, a recent computational study found that dendritic inhibition can lessen dendritic excitability more effectively when inhibitory synapses are located farther from the soma than an excitatory source, "off-path", as opposed to being positioned between the soma and excitation, "on-path" [21]. Because of the dendro-centric viewpoint of this finding, it is not contradictory but rather complementary with studies that show that "on-path" inhibition is more effective at preventing action potential generation at the soma [16,24–26].

To date, investigations into inhibition's dampening effect on dendritic excitability have paid little attention to the inhibitory reversal potential [21], or how this could change over time with continued synaptic drive [26,27]. However, fast dendritic inhibition is predominantly mediated by type A γ-aminobutyric acid receptors ($GABA_A$Rs) and to a lesser extent glycine receptors, both of which are primarily permeable to chloride ions ($Cl^-$) [28–30]. As a result, the inhibitory reversal potential depends on the transmembrane $Cl^-$ gradient, which is a

dynamic variable that can change depending on the balance of Cl⁻ ion fluxes into and out of dendritic processes [26,31] with implications for the effects of GABAergic signalling [32,33]. Previous experimental and modelling studies have shown that because of their small volume, dendrites are particularly susceptible to activity-dependent Cl⁻ accumulation and shifts in the inhibitory reversal potential (EGABA) [27,34–37]. However, how Cl⁻ dynamics and different inhibitory reversal potentials affect dendritic inhibition's ability to control dendritic excitability has not previously been investigated. We used computational models built using the NEU-RON modelling framework to answer the important question, "Where should inhibitory synapses be placed within a branched dendrite to minimise dendritic excitability over time?"

We first extended a metric established by Gidon and Segev [21], which allowed us to quantify the effectiveness of synaptic inhibition of different reversal potentials (EGABA) on nearby dendritic excitatory input. We termed this metric the "Inhibitory Level" (IL). Second, we employed this metric to demonstrate that inhibition accumulates at an encircled branch junction. This inhibitory accumulation is dependent on the number of branches and location of inhibitory synapses but is largely independent of EGABA. Third, we find that hyperpolarizing inhibition accumulates at branch points even for very distally placed inhibitory synapses. This is not the case for shunting inhibition, where there is an optimal distance from a branch junction where inhibitory synapses should be located to maximise inhibitory accumulation. Fourth, we find that adding more inhibitory synapses increases absolute dendritic inhibition, but that the extent of inhibitory accumulation across a tree is maintained. Fifth, by accounting for Cl⁻ dynamics we demonstrate that the Cl⁻ loading that occurs during continuous inhibitory synaptic input erodes the effectiveness of absolute dendritic inhibition. However, this does not reduce the ability for dendritic inhibition to accumulate. As a result, we find that the optimal placement of inhibitory synapses to maximise dendritic inhibitory effectiveness is close to, but not precisely at, a shared branch junction. This placement balances a trade-off between enhanced inhibitory accumulation when inhibitory synapses encircle a branch junction with the deleterious, cumulative effects of Cl⁻ loading when inhibitory synapses are close together. The addition of a Cl⁻ sink emphasises the functional role non-active branches can have on preventing Cl⁻ loading and improving inhibition.

## Methods

Multiple synaptic inputs, together with passive channels throughout the neuron and any current applied externally, caused a change in membrane voltage in each dendritic compartment according to:

$$C_m \frac{dV_m}{dt} = -g_{pas}(V_m(t) - V_{rest}) - I_{syn}(t) + I_{axial}(t) + I_{ext}(t)$$

where $C_m$ is the membrane capacitance (1 μF), $V_m$ is the membrane potential, $g_{pas}$ is the leak channel's conductance (0.05 mS, the inverse of the membrane input resistance, 20 MΩ), $V_{rest}$ is the resting membrane potential (-65 mV), $I_{syn}$ is the sum of synaptic currents, $I_{axial}$ is the sum of axial currents and $I_{ext}$ is externally applied current (0.001 nA). See Table 1 for abbreviations, constants, and parameter values.

Axial currents between compartments j and k were calculated using Ohms law with an axial resistance $R_{axial}$ of 0.1 MΩ · cm⁻¹:

$$I_{axial} = \frac{V_{m,j} - V_{m,k}}{R_{axial}}$$

**Table 1. Symbols, constants, parameters, and variables.**

| | Description | |
|---|---|---|
| **Symbols** | | |
| C | Coulombs unit | |
| $Ca^{2+}$ | Calcium ions | |
| $Cl^-$ | Chloride ions | |
| d | Location of input current, in electrotonic units | |
| Δ | Delta, colloquially meaning difference between or change in something | |
| E | Reversal potential, the value of which there is no net flow of current for that transmembrane channel or ion species | |
| $g_{GABA}$ | Time-varying conductance for $GABA_AR$ | |
| GABA | γ-aminobutyric acid, the neurotransmitter released by interneurons | |
| $GABA_AR$ | GABA type A receptor that mediates fast inhibitory synaptic transmission when GABA is bound | |
| $GABA_BR$ | GABA type B receptor that mediates slow inhibitory synaptic transmission when GABA is bound | |
| Glutamate | Primary neurotransmitter released by pyramidal cells in cortex and hippocampus | |
| $HCO_3$ | Bicarbonate ions | |
| i | Location of inhibitory synaptic input, in electrotonic units | |
| $I$ | Current in amperes (A). Inward current is negative by convention | |
| $I_{Cl^-}$ | Chloride ion current through a channel | |
| $I_{HCO_3^-}$ | Bicarbonate ion current through a channel | |
| $I_{GABA}$ | Total current through $GABA_AR$ | |
| IL or $IL_d^i$ | Inhibitory Level. The impact that an inhibitory synapse at location i would have on an input current at location d. i and d are typically omitted whereby i is fixed and d is varied. Defined as $\frac{V_d - V_d^i}{V_d}$ (see **Variables**). | |
| $IL_0$ $IL_{d = i}$ | Inhibitory Level at the junction of multiple dendritic branches Inhibitory Level at the inhibitory synapse i | |
| $IL_{stat}$ | Inhibitory Level determined with static chloride | |
| $IL_{dyn}$ | Inhibitory Level determined with dynamic chloride | |
| $K^+$ | Potassium ion | |
| KCC2 | Type 2 potassium-chloride cotransporter | |
| $Mg^{2+}$ | Magnesium ion | |
| $Na^+$ | Sodium ion | |
| NKCC1 | Type 1 sodium potassium chloride cotransporter | |
| NMDA | N-methyl-D-aspartate, which selectively activates the NMDA receptor, a mediator of slow excitatory synaptic transmission when glutamate is bound | |
| $Vol$ | Volume of neuronal compartment | |
| X | Electrotonic unit. Proportion of the space constant, which is a measure of how far voltage will travel within a neuronal compartment as it attenuates with distance | |
| **Constants** | | **Value** |
| F | Faraday's constant | 96485 s·A·mol$^{-1}$ |
| R | Ideal gas constant | 8.3145 J·K$^{-1}$·mol$^{-1}$ |
| T | Absolute temperature (= 37˚C) | 310.15 K |
| **Parameters** | | **Default Value** |
| $[Cl^-]_o$ | Extracellular chloride concentration | 135 mM |
| $[HCO_3^-]_o$ | Extracellular bicarbonate concentration | 23 mM |
| $[HCO_3^-]_i$ | Intracellular bicarbonate concentration | 12 mM |
| $\alpha_{GABA}$ | GABA$_A$ receptor binding rate [61] | 5 mM$^{-1}$ ms$^{-1}$ |
| $\beta_{GABA}$ | GABA$_A$ receptor unbinding rate [61] | 0.18 ms$^{-1}$ |
| $D_{Cl^-}$ | Chloride diffusion constant [62] | 2.03 μm$^2$ ms$^{-1}$ |

(*Continued*)

 Chloride dynamics determine the optimal distribution of inhibitory synapses

**Table 1.** (Continued)

| | Description | |
|---|---|---|
| $E_{leak}$ or $V_{rest}$ | The resting membrane potential or reversal potential for the current leak channel; the voltage of the neuron if there was no external current or synaptic input | See $V_m$ |
| $g_{GABAmax}$ | Maximum conductance for a GABA$_A$ receptor | 1 nS |
| $g_{leak}$ | Conductance for the leak current | 0.00021 µS |
| $P_{KCC2}$ | Pump strength of KCC2 to extrude Cl$^-$ along with K$^+$ [26] | $1.9297 \times 10^{-5}$ mA mM$^{-2}$ cm$^{-2}$ |
| **Variables** | | **Initial Value** |
| $[Cl^-]_i$ | Intracellular chloride ion concentration | 7.25 mM |
| $ECl^-$ | Reversal potential for chloride ions at a GABA$_A$R | -78.13 mV |
| EGABA | Reversal potential for a GABA$_A$R | -70 mV |
| $\nabla$EGABA | The EGABA relative to the resting membrane potential, $V_m(t=0)$–EGABA | -5 mV for $IL_{dyn}$ |
| $V_m$ | Membrane potential (voltage) | -65.00 mV |
| $V_d$ | The integral of $V_m$ with respect to resting $V_m$ at location d. $\int_t^{t+\Delta t} V_m(t) - V_m(t=0)dt$ | 0 mV |

A synapse's input (current, $I_{syn}$) was defined according to its activation (conductance, $g_{syn}$) and its driving force (the difference between $V_m$ and the synapse's reversal potential, $E_{syn}$) to drive a current into or out of the neuron:

$$I_{syn} = g_{syn} \underbrace{(V_m - E_{syn})}_{\text{driving force}}$$

Where each term depended on time, *t*. Synaptic input depended on the flow of ions across the neuronal membrane, through the synaptic channel, according to their electrochemical gradients, $E_{syn}$. Synaptic conductances reduce the membrane input resistance, which can elicit "shunting inhibition" in synapses when $V_m = E_{syn}$, and thereby act as a 'sink' for current travelling along its path. For convenience, we defined the difference between the resting membrane potential ($V_{rest}$) and the reversal potential for an inhibitory synapse (EGABA) as

$$\nabla E_{GABA} = E_{GABA} - V_{rest}$$

Note that although this appears similar to the definition for driving force, which changes with $V_m$ and EGABA, $\nabla$EGABA only changes with EGABA. Inhibitory synapses were, therefore, classified as shunting ($\nabla$EGABA = 0 mV) or hyperpolarising ($\nabla$EGABA < 0 mV).

The effects of inhibitory synaptic input are not just local changes in $V_m$, but also the propagation of voltage, given by the cable equation [38], and the spread of lowered input resistance along a neuron [21]. We captured both of these effects in something we term the "Inhibitory Level" (IL): the impact that an inhibitory synapse at a location i would have on an excitatory input at location d. The IL was defined as

$$IL_d^i = \frac{V_d - V_d^i}{V_d} \tag{1}$$

where $V_d$ is the time integral of voltage deflection at location d with **no** inhibition and $V_d^i$ is the time integral of voltage deflection at location d **with** inhibition at location i. More

explicitly,

$$V_d = \int_t^{t+\Delta t} V_m(t) - V_{rest} \; dt$$

where $\Delta t$ is the time window of integration (5 ms in Fig 1, 50 ms otherwise).

Although this formulation captures transient inputs, simulations had a constant injected current, $I_{ext} = 0.001$ nA, as the source of excitation (moved to different locations d), and inhibitory synapse(s) were modelled as persistent conductance-fluctuating current, "gclamp", as in [24–26]. The gclamp mean conductance, $\langle g \rangle$, was 0.001 µS, with a standard deviation of $0.1 \times \langle g \rangle$. Gclamp inhibitory synapses were selectively permeable to both $Cl^-$ and $HCO_3^-$ ions (4:1 ratio). In order to accurately represent $Cl^-$ dynamics and $Cl^-$ loading via $GABA_ARs$, $I_{Cl^-}$ and $I_{HCO_3^-}$ were calculated separately as follows:

$$I_{GABA} = I_{Cl^-} + I_{HCO_3^-}$$
$$I_{Cl^-} = \chi \cdot g_{GABA}(V_m - E_{Cl^-})$$
$$I_{HCO_3^-} = (1 - \chi) \cdot g_{GABA}(V_m - E_{HCO_3^-})$$

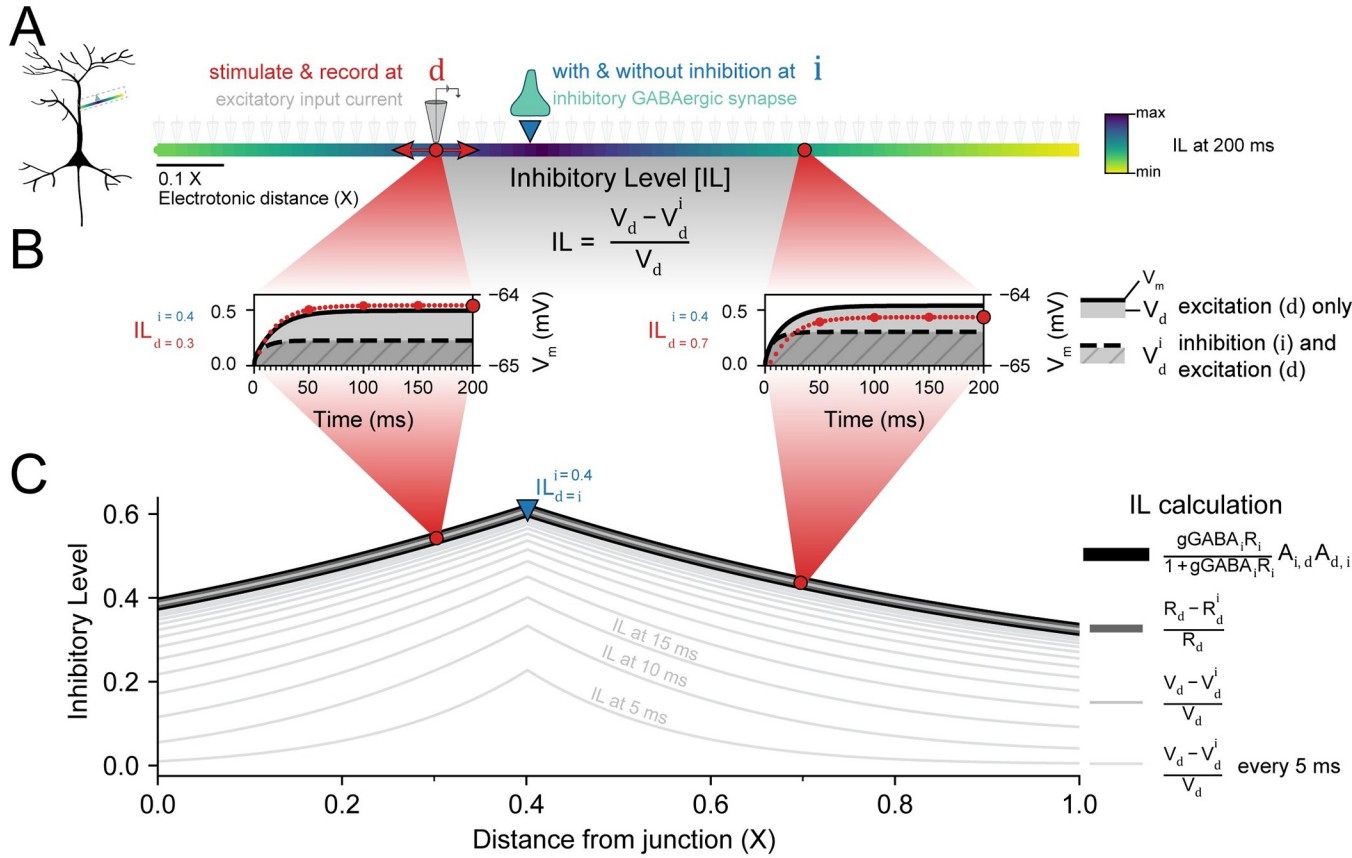

**Fig 1. Inhibitory Level (IL) as a metric to assess the local efficacy of dendritic inhibition.** (A) The effect that an inhibitory synapse at location i (downward triangle) has on an excitatory input current at location d (circle) is termed the Inhibitory Level (IL). The IL at each location d, for stationary i at 0.4 X, is shown for the full length of the dendrite. (B) The IL is calculated by recording the membrane potential with only excitation at d (solid black line) or excitation at d and inhibition at i (dashed black line). The relative difference in the deflection of the membrane potential from rest, between $V_d$ (without inhibition; shaded area) and $V_d^i$ (with inhibition; striped shaded area), is the IL (circles). IL with an integration time window $\Delta t$ of 5 ms for d = 0.3 X (left, red circles) or d = 0.7 X (right, red circles) shows that the steady-state IL is reached within 150 ms. Small circles are IL every 5 ms. Bigger circles are every 50 ms. (C) For shunting inhibition, and given sufficient duration, the numerical calculation of IL (thin light grey line) matches the semi-analytical (medium thickness grey line) and analytical (thickest, black line) solutions. The inhibitory synapse was modelled as a fluctuating $GABA_A$ conductance, $\langle g \rangle = 0.001$ µS, $\sigma^2 = 0.1 \times \langle g \rangle$, and the excitatory input as a constant current, 0.001 nA. $C_m = 1$ µF · $cm^{-2}$, L = 707 µm, r = 1.0 µm, $R_m = 10$ MΩ · $cm^{-2}$, and $R_{axial} = 0.1$ MΩ · $cm^{-1}$.

where $\chi$ represents the fraction of the total GABAergic current carried by Cl⁻ and is given by:

$$\chi = \frac{E_{HCO_3^-} - E_{GABA}}{E_{HCO_3^-} - E_{Cl^-}}$$

where the reversal potential for chloride (ECl⁻) and GABA$_A$ receptor (EGABA) was updated, where applicable, throughout the simulation using the Nernst and Goldman-Hodgkin-Katz equations, respectively:

$$E_{Cl^-} = \frac{R \cdot T}{F} \ln\left(\frac{[Cl^-]_i}{[Cl^-]_o}\right)$$

$$E_{GABA} = \frac{R \cdot T}{F} \ln\left(\frac{\frac{4}{5}[Cl^-]_i + \frac{1}{5}[HCO_3^-]_i}{\frac{4}{5}[Cl^-]_o + \frac{1}{5}[HCO_3^-]_o}\right)$$

where R is the ideal gas constant, F is Faraday's constant, and T is temperature (see Table 1). The values for HCO$_3^-$ were held constant ([HCO$_3^-$]$_i$ = 12 mM, [HCO$_3^-$]$_0$ = 23 mM, and EHCO$_3^-$ = -17.39 mV). Transmembrane Cl⁻ fluxes due to Cl⁻ currents through GABA$_A$Rs, KCC2 co-transporters, as well as changes due to longitudinal/axial diffusion, as modelled in [26], were calculated as

$$\frac{d[Cl^-]_i}{dt} = \frac{I_{Cl^-}}{F \cdot Vol} + P_{KCC2}\left([K^+]_i \cdot [Cl^-]_i - [K^+]_o \cdot [Cl^-]_o\right) + D_{Cl^-}\frac{d[Cl^-]_i}{dx}$$

where P$_{KCC2}$ is the "pump strength" of chloride extrusion (1.9297 x 10$^{-5}$ mA/(mM$^2$·cm$^2$), K$^+$ is the potassium ion, $D_{Cl^-}$ is the diffusion coefficient for chloride in water (2.03 μm$^2$ ms$^{-1}$), F is Faraday's constant, Vol is the volume of the compartment, and x is the longitudinal distance between the midpoint of compartments.

The impact of time-varying EGABA (due to changes in [Cl⁻]$_i$) on the IL was implicitly captured by Eq 1. Unless otherwise indicated, Cl⁻ was static and EGABA was constant throughout any simulation.

Distance was expressed in units of electrotonic distance (X) such that 1.0 X was one space constant (λ). The space constant is a measure of how far voltage will travel within a neuronal compartment as it attenuates with distance [39], and is defined according to

$$1.0\ X = \lambda = \sqrt{\frac{r\ R_m}{2\ R_a}}$$

where r is the radius of the dendrite, R$_m$ is the membrane resistivity, and R$_a$ is the axial resistivity. For a radius of 0.5 μm, R$_m$ of 20 MΩ · cm$^2$, and R$_a$ of 0.1 MΩ · cm, the branch length was 707 μm. All branches were connected to a central compartment 0.01 μm long as in [21].

Simulations were performed with the NEURON modelling framework using the Python interface [40,41]. Full code is available online at https://github.com/ChrisCurrin/chloride-dynamics-and-dendrites.

## Results

### Inhibitory Level as a metric to assess the effect of dendritic inhibition on dendritic excitability

We defined Inhibitory Level (IL) as an extension of the "shunt level" established by Gidon and Segev [21] to account for voltage-dependent effects of hyperpolarising inhibitory synapses (see

Eq 1 above). The IL measures the influence of an inhibitory synapse at location i on an excitatory input at location d. To determine a dendrite's IL for a fixed i (e.g. at 0.4 X), d was varied along the length of the dendrite (Fig 1A). The steady-state IL for a shunting synapse (that is, $\nabla$EGABA = 0 mV) on a single branch was highest at the site of the inhibitory synapse, as found previously [21].

Fig 1B shows the time evolution of $IL_{d = 0.3}$ (left, red circles) and $IL_{d = 0.7}$ (right, red circles) calculated from $V_d$ (solid grey area) and $V_d^i$ (striped grey area). In both cases, the inhibitory synapse location i was 0.4 X. Within 150 ms, IL reached its steady-state value. Because the neuronal membrane is a capacitor and IL is calculated from $V_m$, the IL was taken at 150 ms (using a 5 ms time integration window from 145 ms) unless otherwise stated.

The IL represents a change in membrane voltage deflection caused by inhibition. The IL captures the spatial impact of an inhibitory synapse from both its conductance component, measured as a change of input resistance, as well as its inhibitory post-synaptic potential (IPSP). The IL is, therefore, equivalently.

$$IL_d^i = \frac{R_d - R_d^i}{R_d} - \frac{IPSP}{V_d} \qquad (2)$$

where $R_d$ is the input resistance at location d with **no** inhibition, $R_d^i$ is the input resistance at location d **with** inhibition at location i (see also [42]), and IPSP is the inhibitory post-synaptic potential. The IPSP is zero when a synapse is shunting–its reversal potential is the same as the $V_{rest}$ ($\nabla$EGABA = 0 mV), as in Fig 1. In fact, the IL for shunting synapses can be analytically calculated, as in Gidon and Segev [21]:

$$IL_d^i = \left[\frac{gGABA_i R_i}{1 + gGABA_i R_i}\right] A_{i,d} \, A_{d,i} \qquad (3)$$

where $gGABA_i$ is the conductance of the inhibitory synapse at location i, $R_i$ is the input resistance of the neuron at location i, and $A_{i,d}$ is the voltage attenuation from location i to d or vice versa, which can be calculated using Rall's cable equations [15].

The three methods to calculate IL (Eqs 1, 2, 3) for shunting synapses are equivalent (Fig 1C). However, Eq 3 can be calculated analytically, Eq 2 can be computed semi-analytically using NEURON to record $R_d$ for arbitrary morphologies (along with $V_d$ and IPSP), but Eq 1 needs to be evaluated numerically and taken at the steady-state. Eq 1, however, has two advantages: it works for time-varying input, and it captures both conductance and IPSP effects of inhibition, which is an integral component of hyperpolarising synapses. We therefore used Eq 1 for calculating the IL for the remainder of this study. Note that the conductance and IPSP effects can be disentangled by computing the input resistance along the dendrite and using Eq 2. Furthermore, time-varying input was explored in Gidon and Segev's supplementary material [21]. Here, we focus on frequency-independent phenomena arising from inhibitory synapses.

## Dendritic inhibition combines to suppress dendritic excitability at branch junctions

The IL metric can be easily conceptually extended to the case of multiple branches. First, we consider that multiple branches with inhibitory synapses can be depicted in various ways (Fig 2Ai). Consider, for example, a distal portion of the dendrite with three branches meeting a fourth branch, or equivalently four branches from different parts of the dendrite converging, or the final equivalent structure considered of four nearby branches converging onto a single point (Fig 2Ai). Fig 2Aii shows the visual depictions used throughout this work for 1, 2, 4, 8, and 16 branches with equal number of synapses.

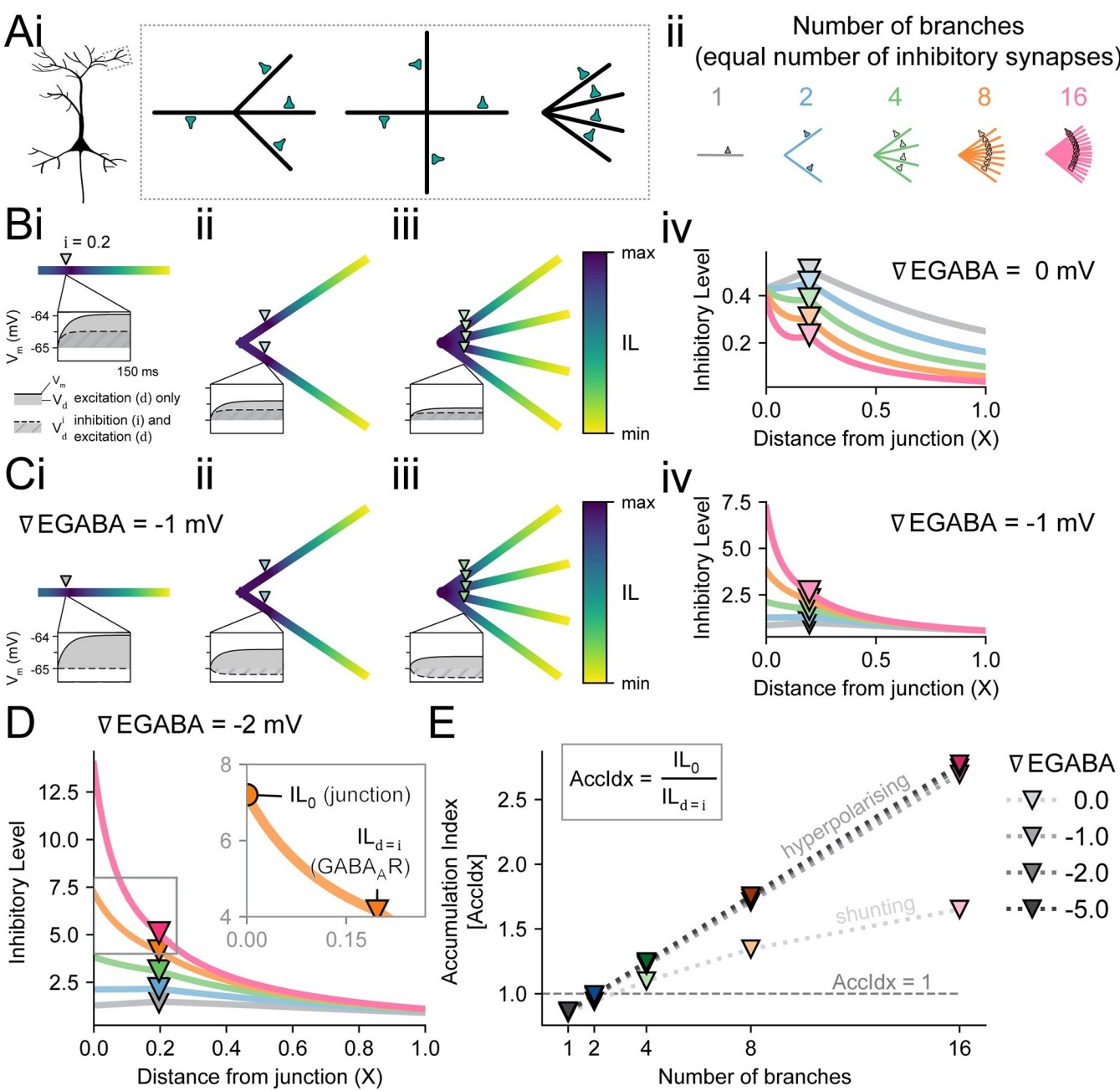

**Fig 2. Dendritic inhibition has a cumulative effect at a dendritic junction.** (A) A 4 branch dendrite morphology can be thought of as 3 branches joining into an identical other branch, all branches coming from different directions, or as 4 branches joining together at a single point. These equivalent situations are pictured in 'Ai', with the rightmost depiction used throughout. The morphologies for different number of branches with equal number of inhibitory synapses are depcited in 'Aii'. (B) The point of maximum Inhibitory Level (IL) for shunting inhibitory synapses at 0.2 X (downward triangle) shifts to the junction as the number of dendritic branches increases, as shown by the heatmaps in 'Bi-iii' and summarised in 'Biv'. The "max" and "min" in the IL heatmaps are the maximum and minimum IL values for each morphology (in contrast to a shared heatmap across all morphologies). (C) When EGABA is set hyperpolarised to the resting membrane potential ($\nabla$EGABA < 0), the inhibitory synapse no longer only shunts (pure conductance-driven effect) but instead also has an additional inhibitory postsynaptic potential (IPSP) contribution. 'Ci-iii' shows the IL heatmaps for 1 (Ci), 2 (Cii), and 4 (Ciii) equal dendritic branches for a hyperpolarising inhibitory synapse at -1 mV below resting $V_m$. 'Civ' shows the corresponding IL values, along with morphologies with 8 or 16 branches. (D) IL for an inhibitory synapse with an $\nabla$EGABA of -2 mV. (D, inset) IL for 8 branches between the junction at 0.0 X and the inhibitory synapse at i (X = 0.2 in this figure). The ratio between the IL at the junction ($IL_0$) and the IL at i ($IL_{d=i}$) is the Accumulation Index (AccIdx) and reflects the cumulative effect of inhibitory synapses on multiple dendrites at a junction. (E) Accumulation Index as a function of the number of branches and the relative reversal potential ($\nabla$EGABA) of the inhibitory synapse (shade of downward triangles).

Using the IL metric, we sought to confirm the finding that inhibitory synapses on multiple branches of a dendritic tree combine so that their cumulative inhibitory effect at a shared dendritic branch junction is greater than at the site of each inhibitory synapse, as demonstrated for the case of shunting inhibition [21]. We found that the functional impact of the inhibitory synapse on an excitatory source, as defined by the Inhibitory Level (IL), was maximised at the junction of a tree with a sufficient number of branches ($> 3$); with each branch having a single synapse at a consistent location (Fig 2). This was the case for both shunting inhibition (Fig 2B) and hyperpolarizing inhibition (Fig 2C).

Given a shunting inhibitory synapse on each branch (at i = 0.2 X), increasing the number of branches decreased the individual influence of each synapse at its own location, while maintaining the same IL at the tree's junction (Fig 2Biv). The IL for a low number of branches ($\leq 2$) was, therefore, highest at the site of the shunting synapse ($IL_{d = i}$; Fig 2Bi and 2Bii). However, once trees with 4 branches (Fig 2Biii) or higher (Fig 2Biv) were considered, then the IL at the junction was higher than at the shunting synapse. This was consistent with the findings of [21]. Also, IL at the junction ($IL_0$) for shunting inhibition remained stable ($\approx 0.4$) for each tree independent of the number of branches. In contrast, for inhibitory synapses with hyperpolarizing reversal potentials ($\nabla EGABA$ = -1 mV, Fig 2C and $\nabla EGABA$ = -2 mV, Fig 2D), IL increased both at the synaptic location, $IL_{d = i}$, and at the shared branch junction, $IL_0$, for increasing numbers of branches (Fig 2C). More generally for hyperpolarizing inhibition, IL at each d increased with more branches and synapses.

Next, we sought to quantify how much IL (inhibitory effectiveness) accumulated at the shared branch junction relative to the IL at the location of the inhibitory synapse. To do so, we created a metric which we termed the "Accumulation Index" (AccIdx) formulated as

$$AccIdx = \frac{IL_0}{IL_i} \tag{4}$$

where the inhibitory synapse location i was typically 0.2 X, as shown in Fig 2D inset. The AccIdx can be intuitively understood as a relative view of dendritic inhibition; specifically, how much more inhibitory the GABAergic synapse is at the junction compared to its insertion location at i.

To understand the influence of reversal potential on the accumulation of inhibitory effectiveness at a dendritic junction, the AccIdx was investigated as a function of the number of branches and inhibitory reversal potential (Fig 2D). When a dendritic tree had more branches (with each having an inhibitory synapse), the AccIdx increased approximately linearly for the case where inhibition was hyperpolarising and clearly sub-linearly when inhibition was shunting. Interestingly, the relative amount of IL accumulation at a junction for a given number of branches was approximately the same regardless of reversal potential when EGABA was hyperpolarising.

In summary, while IL accumulated more at any junction with more negative inhibitory reversal potentials, the relative IL at the junction compared to the IL at the synapse itself (AccIdx) stayed consistent. That is, a more negative reversal potential (with the same EGABA at every GABAergic synapse) increased IL at both the synapse and junction by the same proportion. Increasing the number of branches caused a greater accumulation of IL at the junction.

## Increasing branch occupancy with inhibitory synapses enhances Inhibitory Level at the junction, but complete branch occupancy saturates the relative accumulation of inhibition

To investigate the contribution of each synapse to the overall IL (how much a subset of inhibitory synapses suppresses the excitatory current at any particular location in the dendritic tree), the number of inhibitory synapses was varied while the number of branches stayed the same.

As the "effective number of synapse"–the percentage of branches which have synapses, $\left(100 \times \frac{\text{number of synapses}}{\text{number of branches}}\right)$ – was increased, the IL also increased at both the synapse and junction (Fig 3). Fig 3A and 3B demonstrates a dendrite with 4 branches and a varying number of inhibitory synapses ($\nabla$EGABA = 0 mV). Although having 1 synapse on 4 branches (25% effective number of branches, Fig 3Ai and 3B, thinnest green line) had a strong IL at the inhibitory synapses' branch, the IL attenuation was rapid. That is, the other "silent" or "unoccupied" branches were minimally influenced by the inhibitory synapse.

When the total number of synapses on branches was increased (75% effective number of branches, Fig 3Aii and 3B, thin green line), the branches with inhibitory synapses began to benefit each other mutually and had an increased IL. The combined effect of inhibitory synapses strongly influenced the IL at the synapses themselves ($IL_{d\,=\,i}$), the IL at the junction ($IL_0$), with the combinatory effect spilling over into the "silent" branches. At 100% effective number of synapses, the results were the same as before (Fig 3Aiii and 3B medium green line). When each branch had 2 synapses per location (each at i = 0.2 X, 200% effective number of synapses, Fig 3Aiv and 3B thickest green line), the IL further increased along the dendrites. The same general pattern was evident for different numbers of branches (Fig 3C) and $\nabla$EGABAs (Fig 3D). In addition, when some branches had more synapses than others (e.g. 150% effective number of synapses), the $IL_0$ was naturally shared between them but the IL along the branches differed between those with and without extra synapses due to the total $g_{GABA}$ at i. This led to cases where the AccIdx for branches with fewer synapses were greater than branches with more synapses, yet the branches with more synapses had the greater IL (see Fig 3F inset).

When inhibitory synapses continued to be added, their added inhibitory benefit diminished (Fig 3E). For large effective number of synapses, inhibitory synapses would already be overwhelming excitatory input so that additional inhibitory synapses would hardly affect the outcome ($V_d^i$ for 350% $\approx V_d^i$ for 400% effective number of synapses). Although there was only saturation in $IL_0$ for a large effective number of synapses (Fig 3E), the AccIdx reached its peak at 100% effective number of synapses regardless of the number of branches or $\nabla$EGABA. This maximal value was reached every multiple of 100%, when every branch had the same number of synapses, but only when $\nabla$EGABA < 0 mV. In contrast for $\nabla$EGABA = 0 mV, AccIdx reached its global max at 100% and lower local max values for every subsequent multiple of 100%. Interestingly, 100% for four branches and 50% for eight branches have the same number of synapses (4) but the AccIdx is greater for eight branches when $\nabla$EGABA < 0. This result echoed the previous finding that more branches enhance the AccIdx when $\nabla$EGABA < 0.

For dendrites with an unequal number of synapses per branch (e.g. 150%), the AccIdx was submaximal for branches with extra synapses (Fig 3F, main panel and inset), while the AccIdx continued to increase for those branches with fewer synapses. The branches with more synapses increased the IL along the dendrite, including at the junction, $IL_0$. The branches without extra synapses had a larger $IL_0$ than their own $IL_{d\,=\,i}$, while the branches with extra synapses had a smaller $IL_0$ than their own $IL_{d\,=\,i}$. Regardless, the difference in AccIdx between branches with and without extra synapses is diminished as the effective number of branches increases (Fig 3F, inset). The sharing of a junction for multiple branches with a mixed number of synapses, therefore, benefited some branches more than others in terms of AccIdx. Collectively these results show how differential occupation of branches with inhibitory synapses cumulatively affects local dendritic inhibitory efficacy.

## The location and distribution of inhibitory synapses determine how inhibition accumulates at a branch junction

The simulations presented thus far had kept the inhibitory synapses at a constant location relative to the branch junction. Furthermore, the distribution of inhibitory synapses had been

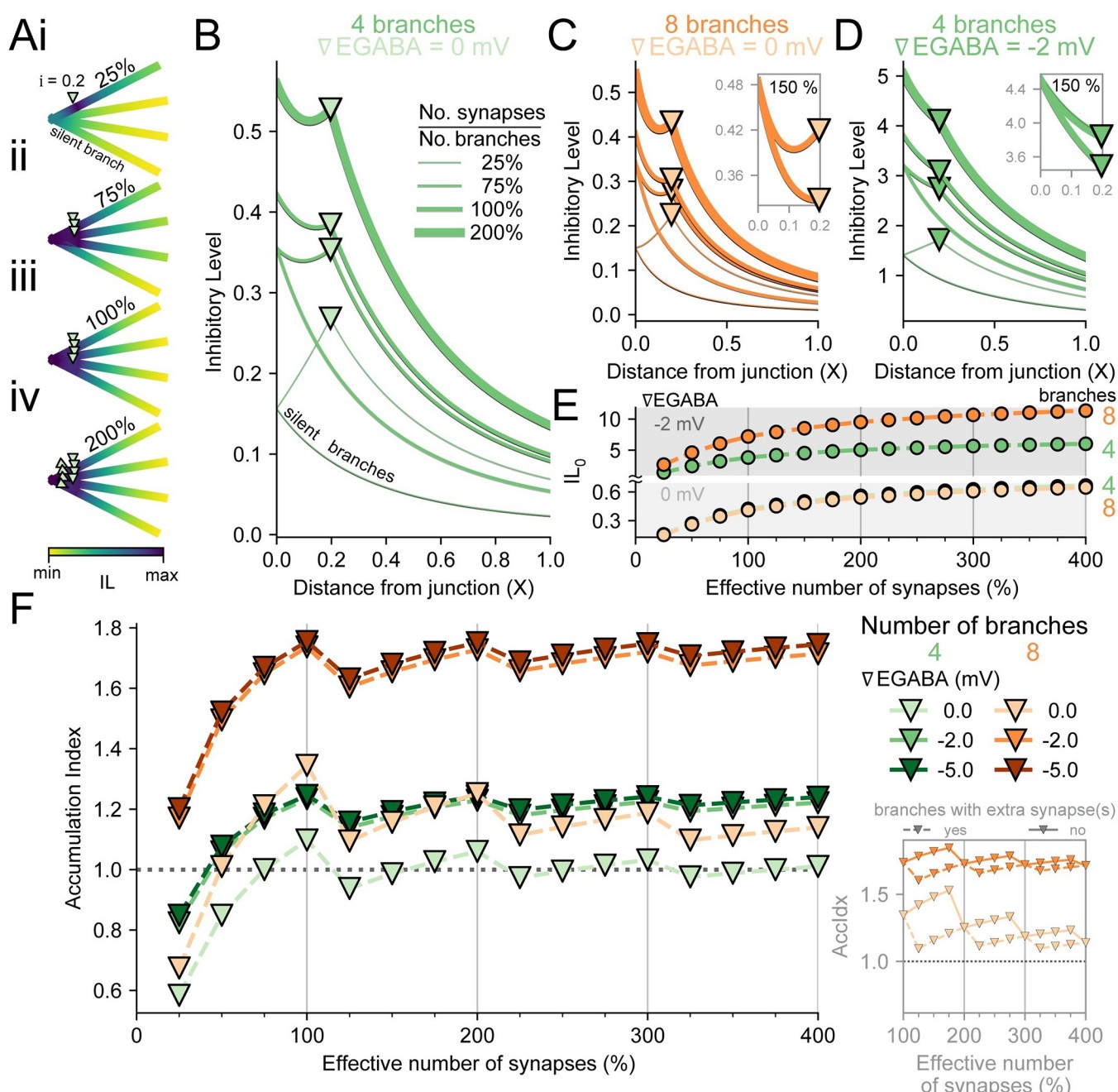

**Fig 3. Increasing branch occupancy with inhibitory synapses enhances Inhibitory Level at the junction but saturates relative inhibitory accumulation.**
**(A)** The percentage of dendritic branches which have inhibitory synapses, "effective number of synapses", indicates the diminishing return in additional IL when adding more synapses. (Ai-iv) 4 branch structure with 1 (Ai, 25%), 2 (Aii, 75%), 4 (Aiii, 100%), or 8 (Aiv, 200%) inhibitory synapses ($\nabla$EGABA = 0 mV). Note that inhibitory synapses are located at 0.2 X, so for 8 synapses on 4 branches (200%), there are 2 synapses per location. **(B)** IL as in 'A' for the effective number of synapses $\left(\frac{\text{number of synapses}}{\text{number of branches}}\right)$. Thinner lines indicate a lower effective number of synapses. Note that there is no "$\nabla$" marker for branches without synapses, "silent branches". **(C)** As in 'B' but with 8 branches. Thus, 75% effective number of synapses for 8 branches is 6 of the branches with synapses and the other 2 without synapses. Inset, 150% (12/8) effective number of synapses has some branches with a single synapse and others with 2 synapses. The branches with extra synapses have stronger IL values while the branches without extra synapses have a moderately better IL than 100% effective number of synapses. **(D)** As in 'B' and 'C', but the synapses on the 4 branches each have $\nabla$EGABA = -2 mV. **(E)** The IL for shunting synapses (lower portion) and IL for hyperpolarising synapses (upper portion) at the junction for 4 and 8 branches with inhibitory synapses either at $\nabla$EGABA = 0 mV or 2 mV. Line and marker colours same as in 'F'. **(F)** Accumulation Index for different dendritic structures when varying the effective number of synapses. Regardless of $\nabla$EGABA or number of branches, the AccIdx is maximised at 100% effective number of synapses. As in Fig 2, $\nabla$EGABA < 0 mV is shifted from $\nabla$EGABA = 0 mV. Note that the number of synapses is equal for 8 branches with 50% effective number of branches and 4 branches with 100% effective number of branches (4 synapses). Inset, the AccIdx for branches without extra synapses (short dashes) are greater than those with extra synapses (long dashes) because these branches utilise the boosted $IL_0$

facilitated by the branches with extra synapses. However, the difference in AccIdx between branches with and without extra synapses decreases with the number of synapses.

consistent across the dendritic tree such that each synapse was at the same distance from the junction ("Tree" distribution). Therefore, we next explored how varying the location and distribution of inhibitory synapses on a branch affected the Inhibitory Level.

To do so, the IL was investigated for different inhibitory synapse locations (placed an equal electrotonic distance from the branches' junction). For a dendritic arbour of 4 branches, shunting inhibitory synapses typically had their maximal IL at the synaptic site itself (regardless of distance from the junction), as demonstrated by example morphologies (Fig 4A, top row, $\nabla$EGABA = 0 mV). In contrast, hyperpolarising inhibitory synapses had their maximal IL at the junction regardless of synapse location (Fig 4A, bottom row, $\nabla$EGABA = -2 mV). Hyperpolarising inhibitory synapses, therefore, more effectively produced cumulative inhibition at sites distal from the synapse locations.

When comparing inhibitory synapse locations for a dendrite with 4 branches, IL at the hyperpolarising inhibitory synapse location ($IL_{d = i}$) increases with proximity to the junction (Fig 4B, bottom). In contrast, $IL_{d = i}$ for a shunting inhibitory synapse is greatest at the junction but weakest near the middle of the branches (Fig 4B, top). Shunting synapses benefitted from branches' terminals by constraining the input resistance attenuation. When voltage attenuation was more of a factor, as in hyperpolarising synapses, then the constrained input resistance was dwarfed by the influence of voltage spread.

As suggested by Fig 4A and 4B, the AccIdx increased with more distal synapse locations when $\nabla$EGABA = -2 mV. In contrast, the AccIdx for $\nabla$EGABA = 0 mV peaked at i = 0.2 X (Fig 4C). Both of these patterns were applicable for more than 4 branches but not 2 and less (Fig 4D). The case for 1 and 2 branches confirmed that these properties only hold once a branching threshold has been reached, which agreed with results from Fig 2D.

To investigate the non-linear effects of synaptic distributions, the "Tree" distribution for 4 branches was compared to a case where all synapses were evenly distributed on a single branch, "Branch", and a case where all the synapses were concentrated at a single location on a single branch, "Focal" (Fig 4E). For $\nabla$EGABA = -2 mV, the maximum IL of the dendritic arbour was different for each distribution. The "Tree" distribution produced the highest IL at the junction (X = 0). The "Focal" distribution predictably had the highest IL at the site of all the inhibitory synapses (X = 0.2), yet interestingly this was still not higher than the IL at the junction produced by the "Tree" distribution. The "Branch" distribution had a broadly raised IL over most of the branch (Fig 4F). The synaptic distributions of inhibitory synapses, therefore, showed different trade-offs in suppressing excitatory input (Fig 4F).

The "Tree" distribution was effective at suppressing the junction the best, which meant inhibitory synapses could be on separate branches a fair distance from the source of depolarisation and yet strongly impacted excitatory current that travelled through the dendrites (Fig 4E and 4F, green and also Fig 2B). However, the suppression of excitatory input farther along the branch (X closer to 1) would be weaker. Extending this to a more realistic pyramidal neuron morphology with a large amount of distal excitatory input and primarily proximal inhibitory input [43], GABA$_A$R synapses on the proximal dendrites can still affect the main trunk even if they are on branches and not directly targeted on the trunk.

The "Branch" distribution had a more uniform IL across its targeted dendrite, while the remaining branches had depressed IL (Fig 4E and 4F, lilac). This distribution would be beneficial if the excitatory source was itself spread on that branch. Finally, the "Focal" distribution

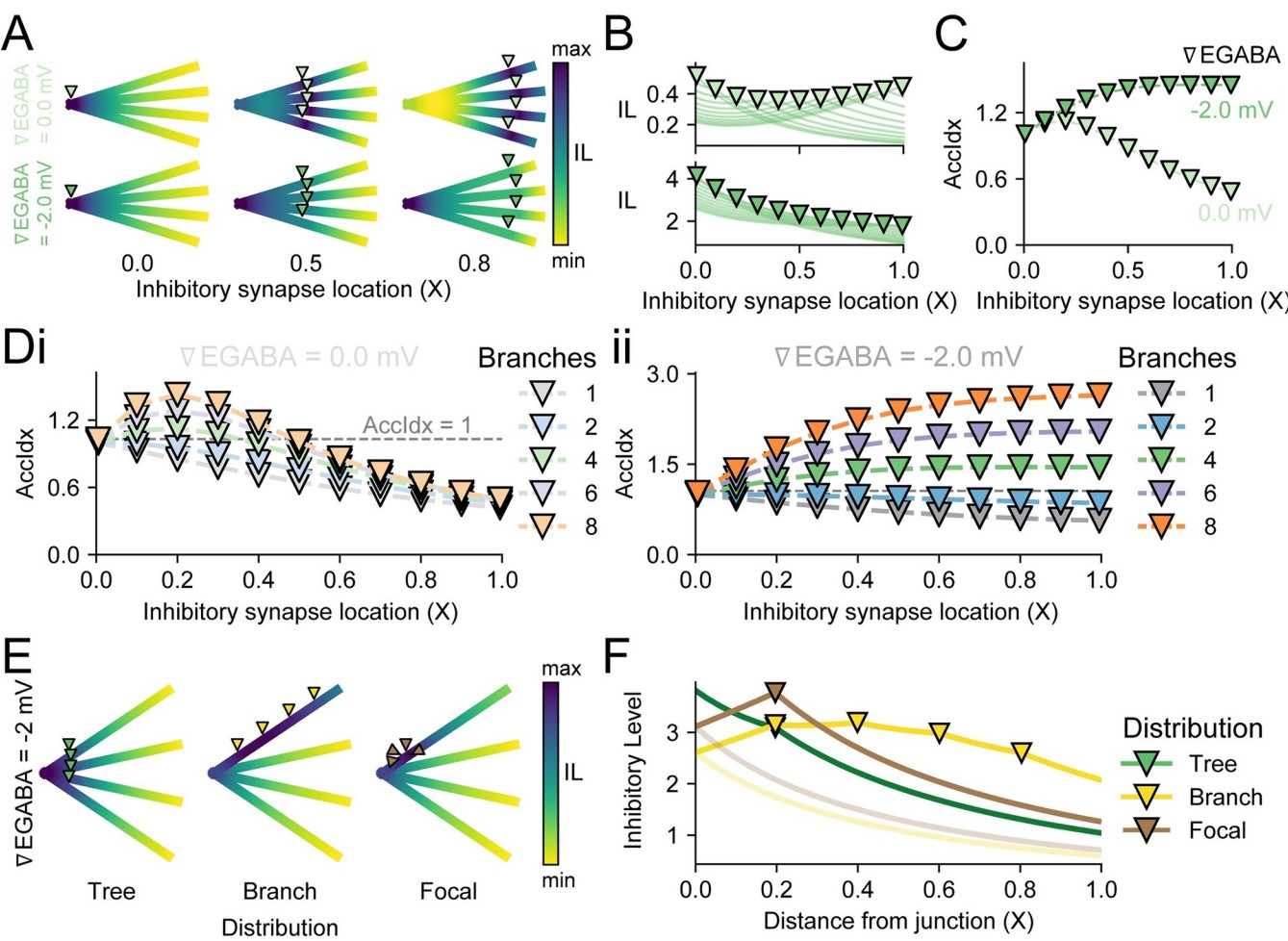

**Fig 4. Location and distribution of inhibitory synapses differentially affect the Inhibitory Level. (A)** Heatmaps show the maximum IL ($\nabla$EGABA = 0 mV, light green markers, top row or $\nabla$EGABA = -2 mV, dark green markers, bottom row) for different inhibitory synapse locations, i, on a 4-branch dendritic structure. The inhibitory synapses are evenly placed from the junction (0.0 X in left column, 0.5 X in centre column, 0.8 X in right column). **(B)** The IL ($\nabla$EGABA = 0 mV, top and $\nabla$EGABA = -2 mV, bottom) for each inhibitory synapse location i, in electrotonic units X, on a 4-branch dendrite. Each trace represents recordings at every excitatory input location D along the dendrite for a given synapse location i. Inhibitory synapses at the junction, i = 0 X, elicit the greatest IL. $IL_{d=i}$ for shunting synapses ($\nabla$EGABA = 0 mV) is the lowest when inhibitory synapses are between the junction and the end of the dendrite (0.4 X), yet $IL_{d=i}$ for hyperpolarising synapses ($\nabla$EGABA = -2 mV) is the lowest at the end of the dendrite (1.0 X). **(C)** The AccIdx for 4 branches with hyperpolarising synapses ($\nabla$EGABA = -2 mV) continues to increase, albeit with saturation, with farther locations for inhibitory synapses. Shunting synapses ($\nabla$EGABA = 0 mV), however, have their greatest AccIdx when i = 0.2 X. **(D)** The different trends in AccIdx between shunting (Di) and hyperpolarising (Dii) inhibitory synapses, as in 'C', holds for dendrites with more than 2 branches. For dendrites with 1 or 2 branches, the AccIdx is greatest when i = 0.0 X. For greater numbers of branches, the maximum AccIdx depends on whether the synapse is shunting or hyperpolarising. **(E)** The maximum IL is dependent on the distribution of the synapses. A dendrite with 4 branches can have 4 synapses placed in different configurations: evenly spaced from the junction on each branch ("Tree"), evenly spaced along a single branch ("Branch"), or all placed at a single location on a single branch ("Focal"). Inhibitory synapses were hyperpolarising ($\nabla$EGABA = -2 mV). **(F)** The IL values for the synapse distributions in 'E'. Although the Tree configuration (green) produces an accumulative IL at the junction, the Focal distribution (turquoise) has the largest absolute IL (at d = i = 0.2 X), and the Branch distribution (lilac) facilitates a more even IL along its branch. However, both the Branch and Focal distributions are branch-selective and hence have to trade-off their gains for weaker ILs on their silent branches (lighter colours).

would maximally target an excitatory source at the same location, but the rest of the dendrite would be left susceptible. The branch selectivity of the "Branch" and "Focal" distributions, therefore, trade-off their gains for weaker ILs on their silent branches (Fig 4F, lighter colours). A neuron's dendrite might use several inhibitory distribution strategies that change over time in response to activity.

## Inhibitory Level is strongly influenced by dynamic chloride over time

Inhibitory current not only depends on a neurotransmitter-driven conductance change but also the voltage-dependent driving force of the inhibitory synapse. Along with time-varying conductance, the driving force can change over time as the membrane potential becomes depolarised by excitatory current and the reversal potential of the $GABA_AR$ (EGABA) increases as $Cl^-$ flows into the cell and is not sufficiently compensated for by $Cl^-$ extrusion mechanisms [44].

The impact of accounting for dynamic $Cl^-$ on the IL was investigated over time for a single branch (Fig 5Ai) as well as 4 branches (Fig 5Aii) with consistent inhibitory location, i = 0.2, initial $\nabla$EGABA of -5 mV, and constant excitatory input as previously mentioned. Along with heatmaps of absolute IL over time for both static $Cl^-$ (where intracellular $Cl^-$ levels were held static, $IL_{stat}$) and dynamic $Cl^-$ (where intracellular $Cl^-$ levels were allowed to fluctuate, $IL_{dyn}$), the difference in IL ($\Delta IL = IL_{stat} - IL_{dyn}$), the relative IL at a time point, and EGABA were also shown in Fig 5A. The heatmaps in Figs 1–4 have used the relative IL, indicated by "min" and "max" instead of absolute values. The development of IL over time was recorded at the inhibitory synapse location ($IL_{d = i}$), i = 0.2 X, for both static $Cl^-$ (Fig 5B, dash-dot lines) and dynamic $Cl^-$ (Fig 5B, solid lines), as well as the $\Delta IL$ (shaded area, same heatmaps as in Fig 5A).

It was immediately apparent that with ongoing synaptic input, the IL of dendrites with dynamic $Cl^-$ decreased compared to the same dendrite with static $Cl^-$ (Fig 5A and 5B). Additionally, a dendrite with dynamic $Cl^-$ took longer to reach its steady-state IL value, which would require $Cl^-$ influx at the synapse to be matched by $[Cl^-]_i$ extrusion mechanisms. In most cases, this had not occurred by the end of a 1 s simulation (Fig 5B). Underlying the difference in IL between a dendrite with static $Cl^-$ and dynamic $Cl^-$ was an increase in EGABA focused at i even though its effect, $\Delta IL$, was more broadly apparent across the dendrite (Fig 5A). Because $Cl^-$ diffuses slowly along the dendrite, compared to voltage propagation, EGABA changes remained relatively localised during 1s of synaptic input (Fig 5A).

We next sought to determine how dynamic $Cl^-$ might affect the relative accumulation of IL at the shared branch junction as compared to the site of inhibitory synaptic input (AccIdx). Although IL is dependent on EGABA, Fig 2D suggests that for negative $\nabla$EGABA, the relative difference between IL at the inhibitory synapse and IL at the junction, i.e. the AccIdx, remains relatively constant. To confirm constant AccIdx during changing $\nabla$EGABA, due to the influence of dynamic $Cl^-$, the $\Delta IL$ at the synapse, $\Delta IL_{synapse}$ (Fig 5C, solid line, left y-axis), and $\Delta IL$ at the junction, $\Delta IL_{junction}$ (Fig 5C, dotted line, right y-axis), were plotted for multiple branches. In each case, although the absolute values were different, the trajectories matched. Considering both changes are in proportion, the AccIdx remains constant over time for each branch structure, after an initial period of capacitance charge (Fig 5E).

Similarly, the change in EGABA was approximately the same for each dendritic structure (Fig 5D). Along with the EGABA heatmaps in Fig 5A, these results indicate that changes in EGABA were relatively localised when i = 0.2 X. Thus, the AccIdx seemed to be independent of EGABA during simulations with dynamic $Cl^-$. However, there was a dependence on the number of branches as previously established (Fig 2D).

Together, these results show that local changes of EGABA at the synapse caused broad depression in $IL_{dyn}$ across the dendrites over time. Although $IL_{dyn}$ changed differently along a dendrite, the AccIdx remained constant. Thus, the accumulation of GABAergic synapses' inhibitory efficacy still occurs even as their absolute efficacy diminishes.

## Dynamic chloride differentially changes the efficacy of dendritic inhibition and dictates optimal synapse distributions

One of the goals of dendritic inhibition is to maximise the suppression of nearby depolarizing excitatory current. For the case of static $Cl^-$, IL would necessarily be greatest when all the

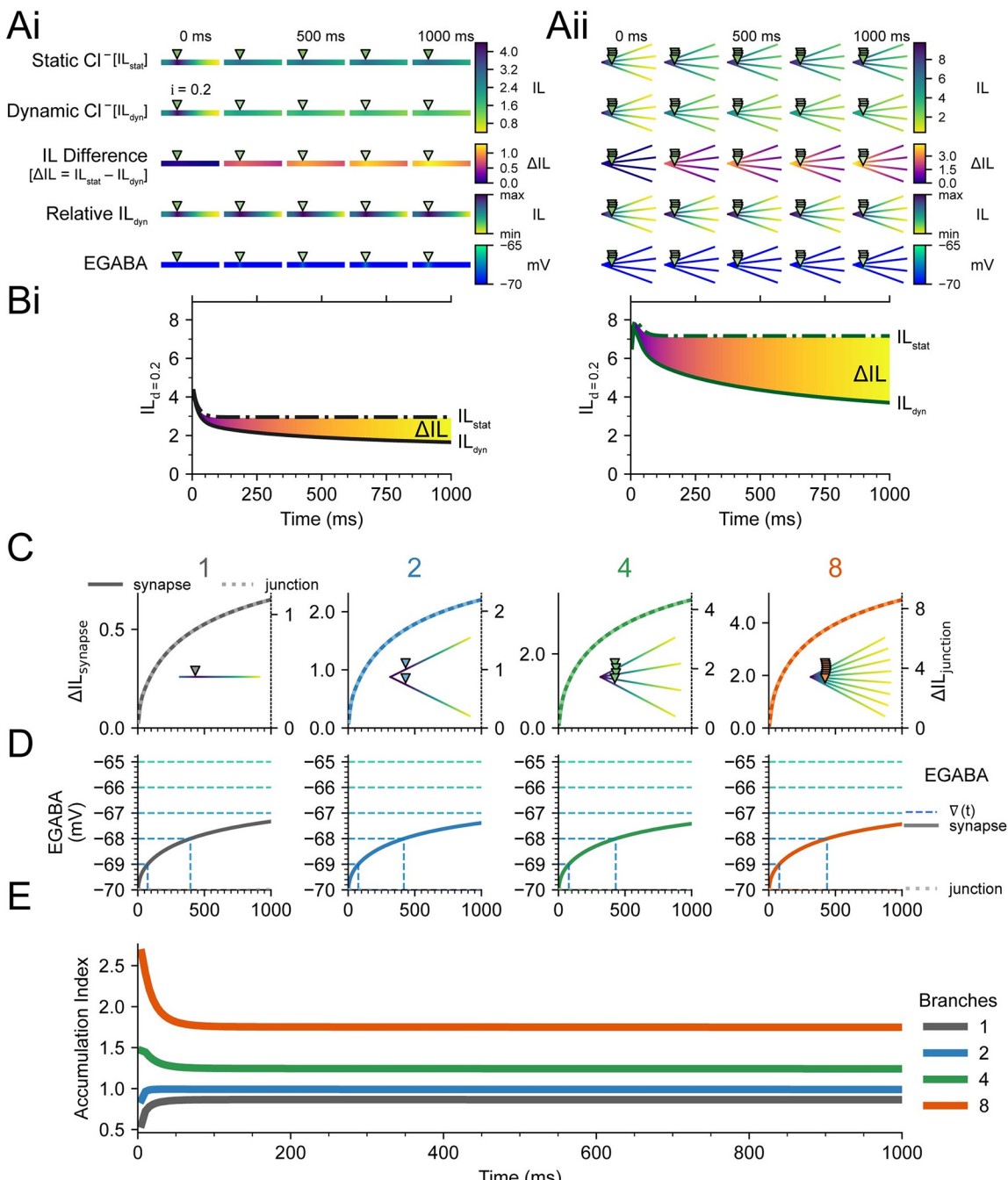

**Fig 5. Chloride loading and shifts in EGABA progressively impact Inhibitory Level, but not Accumulation Index, over time. (A)** IL and related properties calculated at different points in time (5, 250, 500, 750, and 1000 ms), with a backward time integration window, $\Delta t$, of 5 ms. At 5 ms. A dendrite with a single branch, 'Ai', and a dendrite with four branches, 'Aii', is shown. IL with static Cl$^-$ (IL$_{stat}$), and IL with dynamic Cl$^-$ (IL$_{dyn}$), are identical. However, while IL$_{stat}$ reaches its steady-state by the next time point, IL$_{dyn}$ continues to decrease. Note that the heatmap is shared across time as well as IL with static or dynamic Cl$^-$. The difference between IL$_{stat}$ and IL$_{dyn}$, the IL Difference ($\Delta$IL), is strongly focused at the site of the inhibitory synapse but spreads throughout the dendrite over time even while changes in EGABA remain local. The relative IL$_{dyn}$ (IL$_{dyn}$ scaled between that dendrite's minimum and maximum IL$_{dyn}$) indicates that although IL$_{dyn}$ changes over time, the changes are proportional. Initial $\nabla$EGABA was -5 mV (EGABA = -70 mV, $V_m$ = -65 mV). **(B)** IL over a 1000 ms period for both a single-branch dendrite, 'Bi', and four-branch dendrite, 'Bii'. With prolonged input (1000 ms), IL$_{stat}$ decreases until the membrane capacitance is charged and IL$_{dyn}$ stabilises when an equilibrium is reached between Cl$^-$ influx (via GABA$_A$Rs) and efflux (via KCC2). **(C)** The $\Delta$IL at the synapse (solid lines, left y-axes) and $\Delta$IL at the junction (dotted lines, right y-axes) remain in proportion to each other over time, regardless of the number of branches in the dendrite. Note that the scales are different for each axis. These are location-specific traces of what is represented across the dendrite in "Relative IL"

heatmaps. **(D)** EGABA at the synapse (solid lines) increases from -70 mV (-5 mV $\nabla$EGABA) to $\approx$ -67.5 mV over 1000 ms as in 'A'. EGABA at the junction changes only marginally (dotted lines). Vertical dashed lines indicate the time at which EGABA reaches the corresponding horizontal integer values, $\nabla$(t). **(E)** The proportional decrease in IL across the dendrite manifests as a consistent AccIdx, except for during the initial few milliseconds when the membrane capacitance is charging. The AccIdx depends on the number of branches, but not on EGABA or $\Delta$IL.

inhibitory synapses are located at a shared branch junction ($IL_{d\,=\,i\,=\,0}$, Fig 4). However, given the results presented so far, it is not clear that this would remain the case for the more realistic case of dynamic Cl⁻. To determine how dynamic Cl⁻ impacts IL and how this depends on inhibitory location, the IL for co-located excitatory and inhibitory input, $IL_{d\,=\,i}$, was investigated in single and four branched dendrites with varying inhibitory locations, i. Simulations were run for 500 ms with initial $\nabla$EGABA at -5 mV and allowed to vary when Cl⁻ was dynamic (see Methods).

For a single dendrite, the $IL_{d\,=\,i}$ for static Cl⁻ ($IL_{stat}$; Fig 6A, first heatmap row and Fig 6B, inverse triangles) had little deviation and was symmetrical around the maximum IL, which was at X = 0.5. The $IL_{d\,=\,i}$ for dynamic Cl⁻ ($IL_{dyn}$; Fig 6A second heatmap row and Fig 6B, triangles), was lower compared to $IL_{stat}$ for every inhibitory synapse location, and much lower at the ends of the branch (i $\approx$ 0 Fig 6B, left inset, and i $\approx$ 1 Fig 6B, right inset). This drop-off in $IL_{dyn}$ at the ends demonstrates the integral role of Cl⁻ diffusion in reducing Cl⁻ loading: Cl⁻ could only diffuse in a single direction at the branch ends.

For a dendrite with four branches, as demonstrated previously, $IL_{stat}$ was highest at the junction (i = d = 0 X) and decreased with distance from the junction (Fig 6C, first heatmap row and Fig 6D, inverse triangles). However, for the case of dynamic Cl⁻, the $IL_{dyn}$ (IL with dynamic Cl⁻) was highest near–but not at–the junction. This demonstrated how Cl⁻ accumulation weakened inhibition when the synapses were collocated at the junction. That is, Cl⁻ flux through the collocated inhibitory synapses resulted in larger local increases in Cl⁻ concentration and positive shifts in $\nabla$EGABA. It was only when the inhibitory synapses were located a sufficient distance away from the junction and each other (Fig 6D, left inset), i.e. i $\approx$ 0.05 X from the junction, that this was ameliorated and $IL_{dyn}$ reached its maximum value (Fig 6C, second heatmap row and Fig 6C, triangles). The effects of elevated EGABA through Cl⁻ loading when all four synapses were located at, or very close to, the junction were also evident when we plotted the IL difference ($\Delta$IL, the difference between IL with static Cl⁻ and IL with dynamic Cl⁻) as well as heatmaps of EGABA (Fig 6C and 6D). Thus, although $IL_{dyn}$ at or close to a junction may be highest initially (-5 mV $\nabla$EGABA), with continued inhibitory synaptic drive, this $IL_{dyn}$ would become the weakest due to each inhibitory synapse contributing to a highly pooled Cl⁻ load ($\geq$ 0 mV $\nabla$EGABA). Lower left and right insets in Fig 6D further highlight the degraded $IL_{dyn}$ for dynamic Cl⁻ which occurred at junctions and sealed ends (i = 0 and i = 1, respectively).

Our results thus far have indicated a trade-off between maximising IL by having inhibitory synapses at junctions with an initial very negative EGABA, and the cumulative degrading effect that pooled inhibitory input has on the IL due to Cl⁻ loading. The "sweet spot" for an inhibitory location, therefore, lies a small distance away from a junction where Cl⁻ can diffuse away in multiple directions, but inhibitory effects can still accumulate at the shared branch point. This multi-directional diffusion works to help prevent pooling of multiple Cl⁻ currents, which could overwhelm local Cl⁻ extrusion. This "sweet spot" distance from the junction depends on the number of branches, number of inhibitory synapses, strength of inhibition (strongly related to the rate of Cl⁻ influx), Cl⁻ extrusion rate, Cl⁻ diffusion rate, and the size of the neuronal compartments.

Finally, in the context of realistic Cl⁻ dynamics, we specifically explored how different synaptic distributions at different locations over the entire dendrite, might drive the greatest dampening of dendritic excitability, i.e., drive the maximum IL. The only constraint we

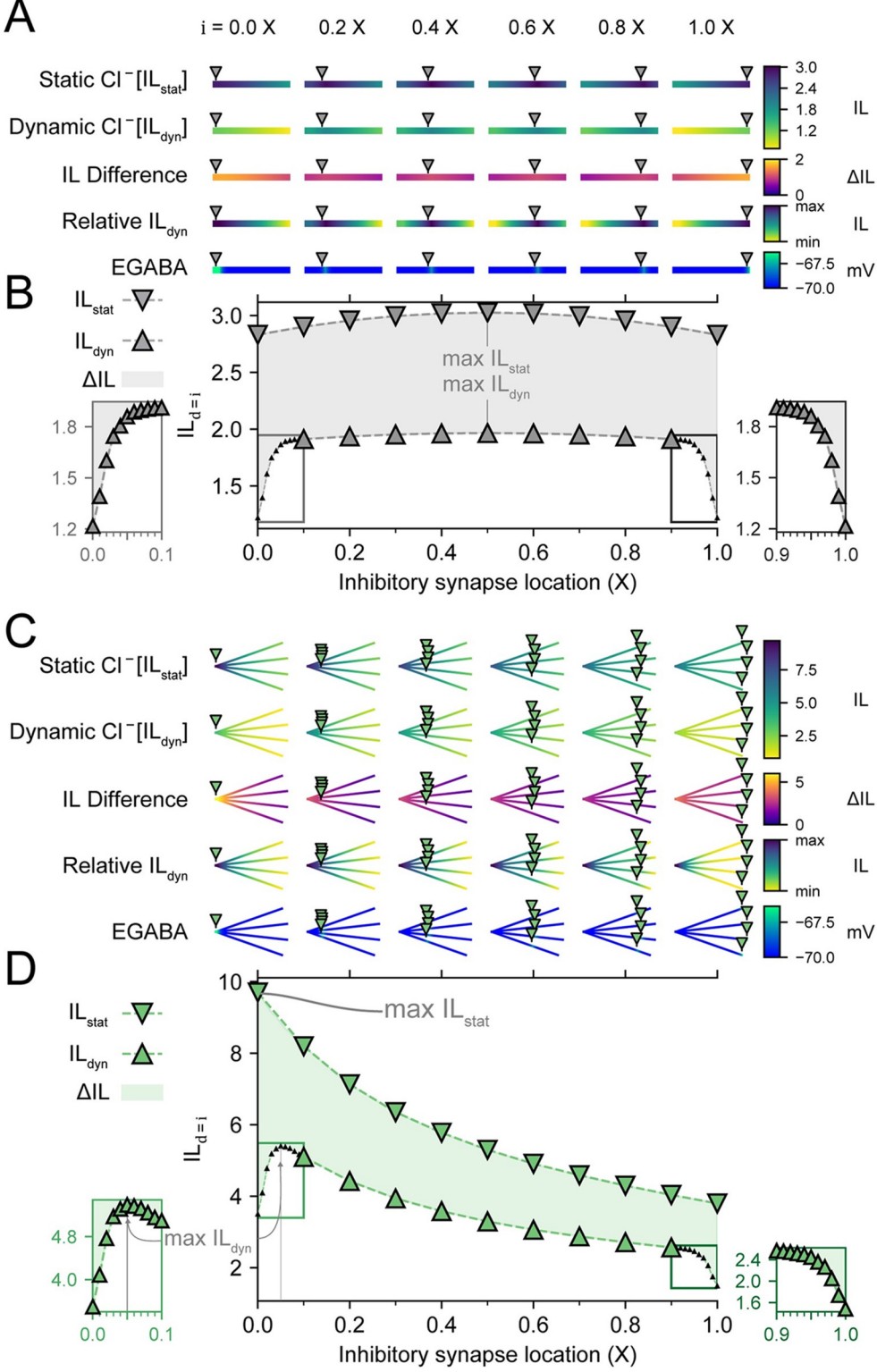

**Fig 6. Dynamic chloride has a differential effect on Inhibitory Level depending on the location of inhibitory synaptic input. (A)** Heatmaps of a single-branched dendrite indicate how IL with static Cl⁻ (IL$_{stat}$) and IL with dynamic Cl⁻ (IL$_{dyn}$), the difference between these (ΔIL) are differentially affected by inhibitory synapse location (0.0, 0.2, 0.4, 0.6, 0.8, 1.0 X). In addition to heatmaps with values shared across synapse location, the relative IL heatmaps are independent of each other and indicate the IL throughout a dendrite scaled to that dendrite's minimum and maximum

IL. Values for EGABA are also represented for each synapse location and indicate local changes in EGABA. All values are taken after 500 ms. **(B)** IL with d at i, $IL_{d = i}$, for both static $Cl^-$ (inverse triangles) and IL with dynamic $Cl^-$ (triangles). The difference ($\Delta IL$) is indicated by the shaded region. Left inset and right inset, when the inhibitory synapse is located near the ends of the branch, each $IL_{dyn}$ is dramatically lower than its $IL_{stat}$ counterpart due to $Cl^-$ loading. **(C)** Same as in 'A' but for a four-branch dendrite. Due to multiple branches sharing a junction, the difference between $IL_{stat}$ and $IL_{dyn}$ are largest at the junction (0.0 X) and are no longer symmetrical around 0.5 X. **(D)** Same as in 'B' but for a four-branch dendrite. Although $IL_{stat}$ is strongest when inhibitory synapses are placed at the junction, $IL_{dyn}$ is weakest in this scenario. Instead, the maximum $IL_{dyn}$ occurs when inhibitory synapses are placed at 0.05 X. Left inset, concurrent activation of multiple inhibitory synapses close to each other causes $IL_{dyn}$ to rapidly decline when inhibitory synapses are close to the junction due to the pooled effects of $Cl^-$ loading. Right inset, the ends of the branches restrict the diffusion of $Cl^-$ and therefore also decrease $IL_{dyn}$.

applied was that although synapses could be placed anywhere, the total number of inhibitory synapses equalled the total number of branches for any given simulation. Put another way, given the same number of synapses as branches, where should these be placed to drive the maximum possible IL in a dendritic tree? We performed this with the three different synaptic distributions: encircling a junction, "Tree", all at the same place, "Focal", or all distributed on a single branch, "Branch".

Fig 7 summarises our findings where the "sweet spot" distance for maximising IL while mitigating $Cl^-$ loading depended on the number of branches and the distribution of the inhibitory synapses (Fig 7A). Because previous results indicated the site of greatest inhibitory effect was either at the inhibitory synapse itself or at the junction, we specifically simulated recording from both locations to see their different responses. As before, dendrites with the "Tree" distribution had their maximum IL at the junction ($IL_0$) when the inhibitory synapses were placed close to, but not at, the junction ($\approx 0.07$ X). With an increased number of branches, the site of maximum $IL_0$ remained the same. The maximum IL at the inhibitory synapses ($IL_{d = i}$) for the "Tree" distribution was reached when the inhibitory synapses were located near 0.05 X. Increased numbers of branches with synapses shifted the maximum $IL_{d = i}$ closer to the junction. Note that for 4 branches, the maximum $IL_{d = i}$ (as in the second "Tree" panel in Fig 7A and in Fig 6D) was at 0.05 X whereas the maximum $IL_0$ (first panel in Fig 7A) was at 0.07 X, which was also the maximum IL overall (as shown in Fig 7B).

If all the synapses were on top of each other, as in the "Focal" distribution, then the $Cl^-$ for all the inhibitory synapses would be concentrated in one area. This focal point worsens $Cl^-$ loading by multiple inhibitory synaptic activation in a limited volume segment with a restricted space for $Cl^-$ to be cleared via diffusion and surface-bound $Cl^-$ extrusion mechanisms. This typically resulted in $\nabla$EGABA becoming positive, especially for many synapses, and negating the inhibitory effect of GABAergic transmission. Spreading the inhibitory synapses evenly along a branch, "Branch" distribution, resulted in broad, but relatively low maximum IL (Fig 7A, rightmost panel).

In summary, encircling a junction ("Tree") was most effective at dampening excitation, with the maximum IL occurring at the junction itself when inhibitory synapses were placed at 0.07 X away (Fig 7B). For a low ($\leq 2$) number of branches, the maximum IL was at the inhibitory synapse itself ($IL_{d = i}$). However, for 4 and greater branches, the maximum IL was always at the junction (i.e. max IL = max $IL_0$). To highlight these placements, they were visualised with both IL and EGABA, relative to that neuron, and the optimal inhibitory synapse locations (downward triangles) and maximum IL (arrowhead) indicated (Fig 7C).

## A parent branch acting as a chloride sink ameliorates chloride loading to shift optimal inhibitory synapse placement towards the junction

In the previous result (Figs 6 and 7), strong $Cl^-$ loading occurred when inhibitory synapses were placed near the junction, and hence near each other. The pooling of $Cl^-$ was also

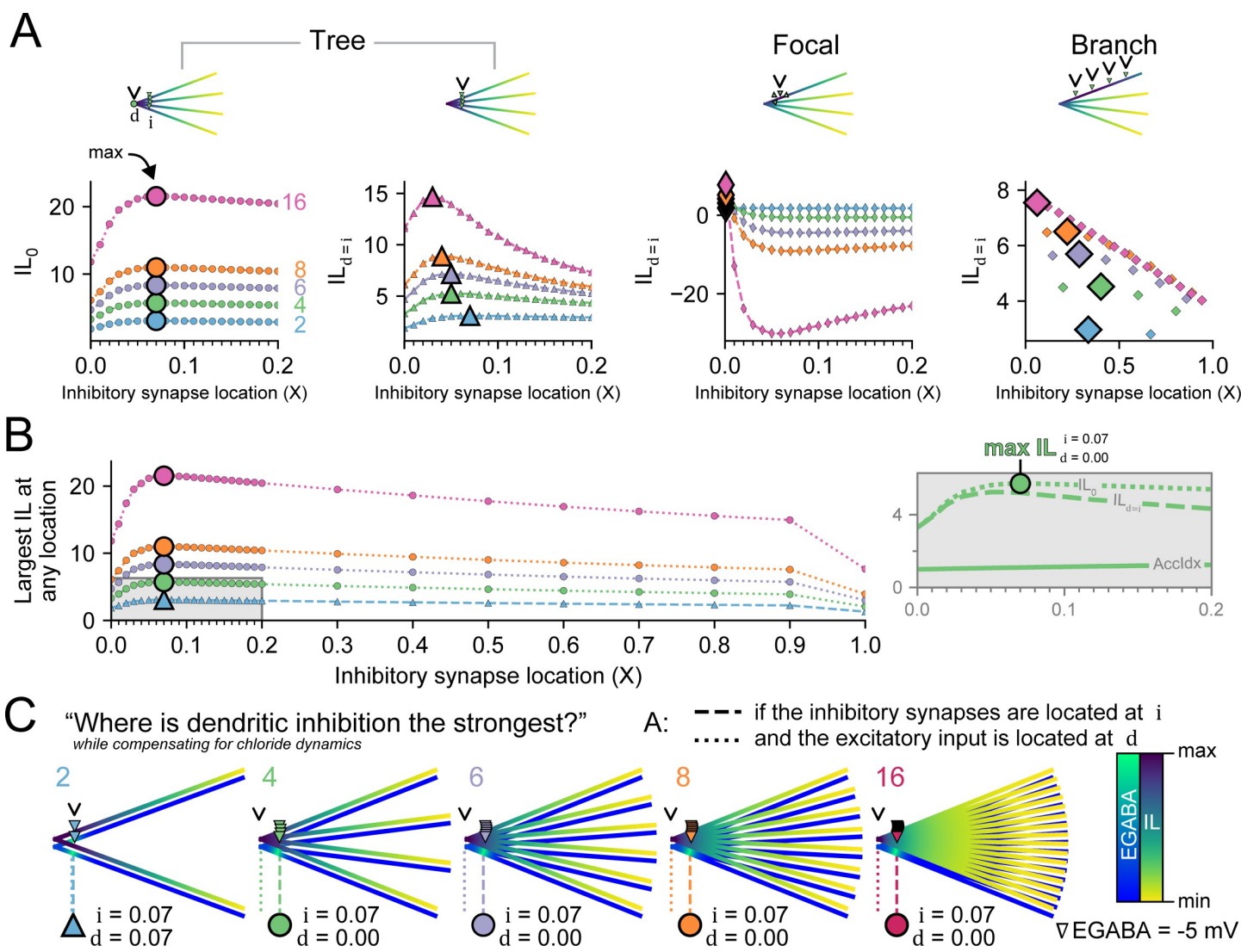

**Fig 7. The optimal placement of inhibitory synapses to maximise the suppression of dendritic excitability.** Given the same number of inhibitory synapses as dendritic branches we determined their optimal placement for different distribution strategies. **(A)** Left plots, for the "Tree" distribution (where each branch has a single inhibitory synapse at location i, example inset), the IL was measured (d) at the junction ($IL_0$) and at the inhibitory synapses themselves ($IL_{d\,=\,i}$). Due to Cl⁻ loading, the maximum IL at the junction ($IL_0$), large marker, occurs when inhibitory synapses are a short distance away from the junction (0.07 X). In contrast, dendrites with more branches generate maximum $IL_{d\,=\,i}$ when the inhibitory synapses are located closer to the junction (0.03 to 0.07 X) as more branches mean that there are more avenues for diffusion to ameliorate deleterious [Cl⁻]ᵢ loading. Second from right plot, Inhibitory synapses placed in the "Focal" distribution (all concentrated at one spot) result in substantial Cl⁻ loading over 500 ms and therefore have an excitatory effect (IL < 0). Right plot, $IL_{d\,=\,i}$ for each inhibitory synapse in "Branch" distributions where inhibitory synapses are placed with even spacing along a single branch. **(B)** The overall largest IL at any location on the dendritic tree is plotted for all varied synapse locations using the "Tree" distribution. The optimal location for inhibitory synapses to create the largest depression of dendritic excitability is ≈ 0.07 X and encircling a junction. For 2 branches (blue), the overall maximum IL is at i itself; but additional branches have their maximum IL at the junction ($IL_0$). This location optimises the cumulative voltage-conductance inhibitory effect of the inhibitory synapses while reducing the pooling of Cl⁻ loading via the synapses themselves. A dendrite with 4 branches and 4 synapses had its maximum IL when the inhibitory synapses were placed at 0.07 X and IL was measured at the junction ($IL_0$). The $IL_{d\,=\,i}$ and the accumulation index (AccIdx = $IL_0$ / $IL_{d\,=\,i}$) are also shown for comparison. **(C)** EGABA and IL heatmaps with the optimal inhibitory synaptic placements (downward triangles) and the location for dampening dendritic excitability the most (arrowhead) for 2 (blue), 4 (green), 6 (purple), 8 (orange), and 16 (magenta) inhibitory synapses and branches.

exacerbated with increasing numbers of branches with inhibitory synapses. In more biologically realistic scenarios, there is often a "sink" that allows excess Cl⁻ to diffuse down its electrochemical concentration gradient. The size of the sink can play a major role in the speed of diffusion, with larger sinks allowing a faster funnelling away from sites of Cl⁻ loading. We,

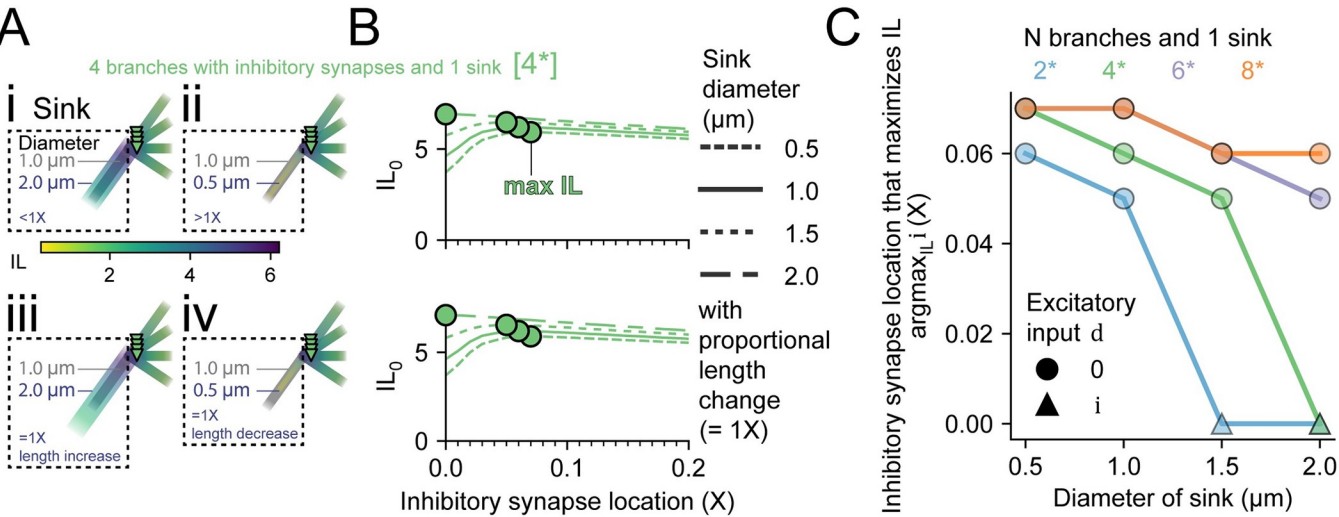

**Fig 8. The optimal placement of inhibitory synapses depends on the diameter of a parent branch, which creates a chloride sink.** A large chloride "sink" means that the inhibitory synapse location that maximises IL (argmax$_{IL}$ i) moves closer to the shared branch junction. **(A)** A parent branch (a "sink") attached to a further 4 branches each with an inhibitory synapse (i = 0.2 X). The size of the parent branch was changed by altering the diammeter from 1 μm (black shade) whilst keeping the length the same, which changes the length constant (<1 X, Ai and >1 X, Aii) or by changing the diameter and length proportionally which maintained the length constant (= 1 X, Aiii and Aiv). The panels Ai-iv use a shared heatmap with the lowest and highest IL across all of them. **(B)** Doubling the sink diameter from 1 μm (solid line) to 2 μm (alternating long dash then short dash) noticeably changed the IL at the junction (IL$_0$) values and moved the max IL (circle marker per diameter) closer to the junction (0.00 X). Halving the sink diameter to 0.5 μm (thinner dashed line) slightly shifted the IL$_0$ trace and max IL was 0.07 X. This pattern, and IL values, were very similar when the parent branch length was changed in porpotion to the diammeter (bottom)). **(C)** Summary panel indicating the effect of sink diameter on the optimal placement of inhibitory synapses for different branch (and synapse) numbers, each having a single sink. Sinks of a larger diameter were more influential in ameliorating chloride-related deterioration in inhibitory efficacy but this amelioration was dampened for more branches. The location of excitatory input d to elicit the maximum IL is indicated by a circle (junction) or triangle (inhibitory synapse location).

therefore, sought to investigate how a parent branch acting as a Cl⁻ sink could influence the inhibitory effectiveness of inhibitory synapses on child branches (Fig 8).

To do so, we added a "parent" branch to the 4-branch structure. Only 4 of the now 5 branches had inhibitory synapses such that parent branch acted as a sink for Cl⁻ (4*, Fig 8). Then, the parent branch diameter was changed from 1 μm (the same as the other dendrites) to either 2 μm (Fig 8Ai and 8Aiii) or 0.5 μm (Fig 8Aii and 8Aiv). In Fig 8Ai and 8Aii, the change in diameter resulted in different length constants, whereas Aiii and Aiv had their lengths proportionally changed to maintain a length constant of 1 X. For a 4-branch structure (Fig 8B), a larger sink diameter helps to recover the deleterious effects of Cl⁻ loading when inhibitory synapses are placed near the junction (note the lack of a sharp drop near the junction). This adjustment shifts the location of inhibitory synapses that elicits the maximum IL (argmax$_{IL}$ i) for 4 branches with 1 sink (4*) from 0.07 X for 0.5 μm sink diameter, to 0.06 X for 1 μm, to 0.05 X for 1.5 μm and 0.00 X for 2 μm (green circles). The curves and maximum IL were virtually identical whether the length was proportionally increased along with the diammeter or not. Finally, we explored how argmax$_{IL}$ i varied with sink diameter for other branch structures (Fig 8C). We found that the influence of the parent branch diameter on the optimal location was heavily dependent on the number of child branches, with 2* and 4* dramatically shifting their argmax$_{IL}$ i from 1 μm to 2 μm compared to 6* and 8*. For static Cl⁻, the argmax$_{IL}$ i would be at the junction, so offsets farther away are due to Cl⁻ loading. These results illustrate how Cl⁻ sinks (branches without inhibitory synapses or the soma) can affect the optimal placement of inhibitory synapses.

## Discussion

Previous work has shown that activity-dependent $Cl^-$ accumulation can compromise the ability of dendritically targeted inhibition to control neuronal output in the form of action potential generation at the soma and axon initial segment [26]. However, it is now well appreciated that dendrites also host active conductances and that non-linear input integration occurs within the dendritic tree itself [45,46]. Therefore, how $Cl^-$ dynamics affect the ability of peripherally targeted inhibition to control dendritic excitatory input is an important issue that remains unexplored. Here we use a metric (the inhibitory level, IL), to quantify the extent to which multiple inhibitory synapses (with variable EGABAs) can control excitatory depolarization within dendritic trees. This allowed us to determine the optimal spatial distribution of inhibitory synapses to maximise local dendritic inhibition. We find that GABAergic synapses with more negative EGABA are multiplicatively better at suppressing local dendritic excitation throughout the branches (increased IL). Extending this, multiple GABAergic synapses can cause greater inhibitory suppression at a shared branch junction than at any of the inhibitory synapses themselves. This agrees with a previous study that demonstrated this effect for shunting inhibition [21]. Interestingly, while absolute IL is increased with more negative EGABAs, the relative ratio of the IL at the junction compared to at the synapses themselves (i.e., the accumulation of inhibitory effectiveness), is constant regardless of EGABA. This suggests that although EGABA sets the strength of inhibition itself, the number of branches and their occupancy by GABAergic synapses sets the relative accumulation of inhibition at a shared branch point.

If EGABA is considered a static variable, the most powerful absolute local inhibition will always be generated if all available inhibitory synapses are placed at the same location; for example, at a branch junction or at the same location of a dendritic branch. In reality, however, EGABA is susceptible to incoming $Cl^-$ currents through $GABA_A$Rs. This is because synaptic $Cl^-$ currents, if large enough, can overwhelm local extrusion $Cl^-$ mechanisms and reduce the transmembrane $Cl^-$ gradient [31]. Our results clearly illustrate the detrimental effects of clustering all available inhibitory synapses at the same location. The intuitive reasoning is that local $Cl^-$ diffusion and transmembrane extrusion may be overwhelmed by the cumulative effect of $Cl^-$ currents via multiple GABAergic synapses at the same location. We also show that this effect is ameliorated to an extent by the presence of $Cl^-$ "sinks", such as large volume parent branches devoid of inhibitory synapses. Nonetheless, as demonstrated here, the negative effects of $Cl^-$ loading could be part of the reason that inhibitory synapses cluster less on branches than their excitatory counterparts [43,47,48].

When accounting for dynamic $Cl^-$, temporal factors like duration and frequency of input become an important consideration for understanding the effectiveness of dendritic inhibition [26,27]. For example, a low-frequency burst or very short duration of inhibitory synaptic input will result in negligible $Cl^-$ concentration changes. Therefore, in comparison to the analysis presented here which used 1s of continuous inhibitory input, the optimal placement of inhibitory synapses to maximise IL would be closer to a shared junction. However, high-frequency bursts or sustained inhibitory inputs can drive substantial changes in $[Cl^-]_i$ that, as we have demonstrated here, strongly affect IL. Models of branched dendritic trees have previously been conceptually and mathematically reduced to a single large cylinder with the same inputs [19,49]. It is important to note that this approach of equivalent cylinders does not hold for the case of dynamic $Cl^-$, as the precise morphological structure (volume, surface area, spines, tortuosity, etc.) and compartmental ion differences dictate the kinetics of $Cl^-$ dynamics and consequent effects on GABAergic inhibition.

Given the reality of Cl⁻ dynamics in dendrites, our simulations predict that the optimal distribution of available GABAergic synapses to maximise local inhibition in a branched dendritic tree is to place synapses surrounding, but not at, a branch junction. This allows centripetal accumulation of inhibitory effectiveness while minimising the detrimental effects of Cl⁻ loading that occurs when GABAergic synapses are all placed at the same location. Indeed, experimental evidence suggests that dendritic inhibition (typically from somatostatin-expressing–SOM⁺–interneurons) is widely distributed across pyramidal cell branches instead of clustering synapses on a single branch, or primary dendrite [12,43,50–52]. This strategy has several advantages. First, inhibitory synapses do not need to control excitation by directly targeting each excitatory synapse. Second, dendritic inhibition targeting a particular branch could control local integration there [52,53], but activating widespread inhibitory synapses across multiple branches would be able to accumulate at a shared-branch junction, or primary dendrite, to gate the generation of dendritic action potentials and the associated burst firing of pyramidal neurons [54]. Third, this spatial arrangement would minimise the deleterious effects of Cl⁻ loading with continued synaptic drive and the accompanying potential for dendritic inhibition to fail, which would otherwise facilitate the onset of seizure activity [55]. Finally, branches without active inhibitory dendrites can still serve a functional role by acting as Cl⁻ sinks for branches with active inhibitory synapses.

In our models we did not incorporate non-linear conductances (such as NMDA receptors, voltage-gated $Ca^{2+}$, $Na^+$ or $K^+$ channels as well as hyperpolarization-activated cation currents) and therefore did not model their interactions with, and possible effects on Cl⁻ loading and inhibition in dendrites. As described above our work predicts that spatially distributed dendritic inhibition, by minimising Cl⁻ loading whilst still allowing centripetal accumulation of inhibitory effectiveness would more powerfully prevent the activation of depolarization-gated $Ca^{2+}$ and $Na^+$ currents and NMDARs than spatially localised inhibitory synapses. It should be noted that the activity of excitatory voltage-gated cation currents themselves, by increasing the driving force for Cl⁻ influx during concurrent inhibitory input, would serve to accelerate Cl⁻ loading and more rapidly erode the strength of inhibition. The possible effects of HCN channels and voltage-gated $K^+$ channels with inhibition in dendrites are more difficult to predict. Nonetheless it is worth noting that these non-linear conductances can have paradoxical effects on synaptic input [56].

Our approach utilised the equivalent circuit, charge sum approach to modelling the dynamics of the membrane potential in dendritic compartments using the well-established NEURON simulation environment [63]. This entailed computationally tractable but simplified ohmic formulations of transmembrane and axial ionic currents. Furthermore, axial diffusion of Cl⁻ between compartments was modelled as a diffusion process as opposed to employing electrodiffusion, which is thought to be particularly important at small spatial scales such as dendritic spines [57,58], which were not modelled here. With increases in the availability of computational resources, future studies could attempt to confirm our findings in more realistic dendritic trees, which include dendritic spines and by explicitly incorporating electrodiffusion.

Experimentally verifying the local effects of dendritic inhibition is incredibly challenging [11]. However, continually advancing methods allowing for more precise control of dendritic circuits, e.g. dendritic patching [59] and spatially controlled 2-photon uncaging of glutamate and GABA [60], could lead to the potential for direct experimental verification of our predictions for the optimal placement of GABAergic synapses to control local excitation. Taken together, our modelling work using simple dendritic structures provides a framework for understanding the optimal distribution of GABAergic synapses to maximise suppression of dendritic excitability.

## Acknowledgments

The authors would like to thank Albert Gidon for our discussions on the shunt level, and Kira Dusterwald for their input on the manuscript.

## Author Contributions

**Conceptualization:** Christopher Brian Currin, Joseph Valentino Raimondo.

**Data curation:** Christopher Brian Currin.

**Formal analysis:** Christopher Brian Currin.

**Funding acquisition:** Joseph Valentino Raimondo.

**Investigation:** Christopher Brian Currin.

**Methodology:** Christopher Brian Currin, Joseph Valentino Raimondo.

**Resources:** Joseph Valentino Raimondo.

**Software:** Christopher Brian Currin.

**Supervision:** Joseph Valentino Raimondo.

**Validation:** Christopher Brian Currin, Joseph Valentino Raimondo.

**Visualization:** Christopher Brian Currin.

**Writing – original draft:** Christopher Brian Currin, Joseph Valentino Raimondo.

**Writing – review & editing:** Christopher Brian Currin, Joseph Valentino Raimondo.

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
