## [Decision Letter · Decision Letter 0]

7 Jan 2022

Dear Dr Currin,

Thank you very much for submitting your manuscript "Computational models reveal how chloride dynamics determine the optimal distribution of inhibitory synapses to minimise dendritic excitability" for consideration at PLOS Computational Biology.

As with all papers reviewed by the journal, your manuscript was reviewed by members of the editorial board and by several independent reviewers. In light of the reviews (below this email), we would like to invite the resubmission of a significantly-revised version that takes into account the reviewers' comments.

I feel that it is important to add some additional comments about this decision:  Often, papers submitted to this journal are reviewed by three referees.  In this case, two reviews came back before a third reviewer agreed to complete a review.  Because both reviews are quite thorough and perceptive, I am comfortable moving forward without an additional review and have made the decision to do so.  

Both reviewers express significant enthusiasm for the general project and aspects of the approach but also raise a significant number of major, very important concerns.  In my opinion, these concerns will take substantial work to address, to the point where I had to think carefully about whether a more appropriate path for this manuscript would be rejection (i.e., since a new manuscript that incorporates the additional work could be submitted to this or another journal in the future).  But, because of the positive aspects and the good fit with these reviewers, I have decided that allowing for major revisions and resubmission is the right course of action.  Assuming that you complete these steps, as I hope you will, the same reviewers will be asked to review the revised paper, and the level of satisfaction that they express about the revisions will play a major role in determining the outcome at that stage.  In other words, to use the usual PLOS phrasing, we cannot make any decision about publication until we have seen the revised manuscript and your response to the reviewers' comments. 

Sincerely,

Jonathan Rubin

Associate Editor

PLOS Computational Biology

Kim Blackwell

Deputy Editor

PLOS Computational Biology

Reviewer's Responses to Questions

**Comments to the Authors:**

Reviewer #1: In this study, Currin et al., characterize how the location of GABAergic synapse on dendritic branches impact the ability of these synapses to regulate dendritic excitability quantified here by the metric termed “Inhibitory Level”. A major strength of this study is that it incorporates variable strengths of the GABAergic reversal potential and chloride dynamics which strongly impact GABAergic synapse. Overall, I am enthusiastic about this work, however, there are some concerns that should be addressed.

Major Concerns

1. The schematic in Fig. 1 A is confusing and makes it hard to understand the setup of the simulated experiment. Because there are two electrodes drawn on the schematic, one darker than the other, and only one labeled with “excitatory input current” it appears that there is an excitatory input at 0.7X and a recording at 0.3X that is moving around to measure IL.

2. In the neuron schematic in Fig.1 A shows a somatic compartment connected to the left side of the dendrite however this is omitted from the schematics of the dendritic trees in the following figures. It is also unclear from the methods section if in your simulations the branch point (left end points) of your dendritic trees are capped or if they are connected to a somatic compartment or another dendritic compartment. The branch point should be connected to an additional compartment representing the soma or an additional dendritic compartment. Please clarify as these features will strongly impact model behavior such as the accumulation index and inhibitory level.

3. In the introduction and discussion, the authors point out that dendrites possess active properties driven by NMDA, voltage-gated calcium, and voltage-gated sodium channels that add to the complexity of dendritic processing. This list should include voltage-gated potassium currents and hyperpolarization-activated cation currents. Inclusion of these additional currents is beyond the scope of the current study. However, it should be discussed how inclusion of these currents could impact the results of this study since the interaction between nonlinear conductances have been shown to have paradoxical effects on synaptic inputs (George, M. S., Nature neuroscience 2009).

4. The finding that “multiple GABAergic synapses can cause greater inhibitory suppression at a shared branch junction than at any of the inhibitory synapses” is interesting and consistent with the effects demonstrated by Gidon Neuron 2012. Giddon et al. also showed that distal “off-path” rather than proximal “on-path” inhibition was more powerful dampening proximal excitable dendritic inputs and thus powerfully impacted the postsynaptic neurons output. However, in the current study, the inhibitory level on the proximal end of the dendrite (0.0 X) is the lowest when the inhibitory inputs are on the distal end of the dendrite (1.0 X), Fig.4B. These differences should be discussed.

5. It should be discussed how these results may apply to glycinergic inhibition which should be less sensitive to Cl- fluctuations since these synapses do not flux HCO3 and therefore are more hyperpolarized. Should the distribution of glycinergic synapses be distributed differently than GABAergic synapses to optimize the dendritic inhibitory level?

6. Why was the GABAergic reversal potential not calculated with the Goldman-Hodgkin-Katz voltage equation which is standard for calculating reversal potentials that are permeable to multiple ions?

Minor Concerns

1. Fig. 1C. It is difficult to tell that there are three overlapping curves in C and the light gray dashed lines that show IL every 5ms are very hard to see.

2. In Fig.3 are the curves in panel E supposed to be color coded with panels B-D?

3. In Fig.3 the term “effective number of synapses” in the caption but “effective number of branches” in the figure labels.

4. Figure 4. Is the label “Inhibitory level” that is between the color bar for panel A and the x axis in panel B a shared label?

5. Fig.4 F. The colors are a little hard to distinguish between the curves for the tree, branch and focal distributions.

6. Doyon et al., Frontiers in Cellular Neuroscience 2016, and Phillips et al., eLife 2020 also looked at how Cl- dynamics impact GABAergic inhibition and should be cited.

Reviewer #2: In their manuscript entitled “Computational models reveal how chloride dynamics determine the optimal distribution of inhibitory synapses to minimise dendritic excitability” Christopher Currin and Joseph Raimondo presented a study which ought to identify the optimal distribution of inhibitory synapses within dendritic trees. The authors use a purely computational approach with a simple geometric topology and passive membrane properties. Their main results are that 1.) for 4 and more dendrites with equidistant synapses the local inhibition maximizes at the junction of the dendrites if more that 50% of the dendrites have active inhibitory synapses, that 2.) the level of inhibition superimposes at these junctions with a higher effectivity for hyperpolarizing inhibitory synapses, that 3.) for hyperpolarizing synapses a lower occupancy with active synapses is required for a junctional superimposed inhibition, and 4.) that under consideration of dynamic EGABA the optimal position of inhibitory synapses is about 6-9% of the dendritic length distally to the junction (depending on the number of dendrites).

The question how GABAergic synaptic inputs interacts within a dendritic tree is an important question. The approach to address this question by using a passive dendrite to isolate the general principles of dendritic integration within a dendrite is a reasonable methodology. This study is a significant expansion of basic principles published by Gidon & Segev (2012) and as such can provide new important information about the basic principles underlying the interaction between structure and function of neurons. The implementation of static and dynamic EGABA to identify the optimal localization of inhibitory synapses provides important additional information to comprehend the strategic localization of synapses for efficient inhibitory control. In this respect the use of the simplified models is required to understand the basic principles underlying the principles of interaction between inhibitory synapses. The models and each experimental step are clearly described and nicely illustrated. In particular, I appreciated how the authors linked in the discussion their theoretical assumptions with recent data on the microanatomical distribution of GABAergic inputs.

However, I my opinion in particular the simple geometric structure of the used dendrite model critically limits the transfer of the main finding to a more realistic situation. The limitations of the simplified models, which as mentioned is an important aspect of this study, may be addressed by some exemplary simulations and must be discussed in detail. And finally, there are some small inaccuracies that should be corrected. A consideration of these point may enhance the applicability of this excellent manuscript.

Major:

1.) Topology of the dendritic aggregates:

I have severe concerns that the construction/implementation of your dendritic aggregates may underlie to some extend your main finding that “… the optimal distribution of available GABAergic synapses to maximise local inhibition in a branched dendritic tree is to place synapses surrounding, but not at, a branch junction (Line 514-516). In line 421 you state that “.. each inhibitory synapse [contribute] to a highly pooled Cl- load” thereby enhancing the positive shift of EGABA. However, this observation was most probably caused by the rather artificial topology of your dendritic aggregate, with n dendrites of equal function. Only in this case four GABA synapses can co-localize incrementally close to a common junction. Figure 4E of the Gidon & Segev (2012) depicts the geometric organization of your dendrite arrangement probably more honestly. With this picture in mind, in particularly the fast breakdown of EGABA for synapses close to the junction became more obvious. Regarding the normal dendritic topology, this is, however, a rather artificial situation, as in your case 4 equal dendrites merged without an additional "main dendritic branch" connecting them to the soma (and allowing electrotonic spread and, more important here, diffusion of Cl- ions). Therefore, I would strongly suggest to investigate whether the “optimal inhibitory synapse position” was still observed at the branch point if a diffusional Cl- sink exists in one of the dendrites (which has no synapse).

2.) Quantification of the inhibition level:

At the first sight it was not clear to me, how you can get an IL of 7.5 in Fig. 2Bv. An IL of 1 meant (according to your definition), that the inhibitory synapse eliminated all "excitatory" membrane responses. As soon as you shift to a sufficient hyperpolarization the voltage integral of the excitation+inhibition become smaller (i.e. more negative) and the control integral and thus the numerator in formula #1 becomes larger than the denominator. However, for a IL level of 7.5 the absolute voltage integral with inhibition must be 8.5 times larger than for the control interval (with only excitation). But from Fig. 1B I assume that you depolarized potential is in the range of ca 1 mV. However, if the inhibition abolishes all depolarizing EPSPs and clamps Em to ECl, this means that Em can be maximal 1 mV hyperpolarized to the RMP (resulting in an IL of 2). The most probable explanation I could find is that the total Ri of the dendrite aggregate decreases which each added dendrite, leading to diminished Vd amplitudes under control conditions (which will make the ratio between the GABAergic hyperpolarization and the control depolarization larger). I would suggest to illustrate this interpretation (if it is true) with the voltage responses obtained in multi dendrite aggregates. I wonder whether this diminished voltage response can/must be compensated by increased injection current (e.g. to make the “accumulation index” comparable between conditions with different numbers of dendrites). At least you should discuss how to disentangle the “passive inhibitory effect” from the more elaborated dendritic tree, from the amount of synaptic inhibitory efficacy.

3.) Calculation of EGABA:

The formula provided in line 145 is wrong and cannot be used to calculate EGABA. You have to use the GHK. You can easily test this, eg. with an [HCO3-]i of 12.165481 mM (providing an EHCO3- of -18 mV at 310.15 K and a [HCO3-]e of 24 mM) there is a clear deviation between the GHK-Potential and your simplified formula (e.g. -72.9 vs -71.0 mV at 5 mM Cl; -60.1 vs -56.2 mV at 10 mM Cl-; -51.5 vs -47.7 mV at 15 mM Cl-). Note that the GHK can be interpreted as the "contribution" of the HCO3- permeability depending on the Cl- concentration. Please see Farant & Kaila (2007) for a detailed description of this relation.

4.) Restriction to stationary current:

One conceptual problem I have with your study is the IL was determined by stationary currents and thus the temporal aspects, in particular the effects of gGABA on the membrane time constant, are neglected. Can you at least speculate how this parameter will affect the optimal localization of inhibitory synapses?

5.) Relevance of the first results subchapter

To be strict, I do not really see how several details of this subchapter (Line 169-200) contributes to the overall results of the manuscript. You may consider to eliminate most algebraic terms from here and just define the IL metrics (in accordance to formula #1). In line 198 you state that “Eg. 1 […] captures both conductance and IPSP effects of inhibition”, however, this formula in the present form just includes voltage differences. In addition, I have a problem with the statement “Eq. 3 can be calculated analytically” (line 195). If I understand all parameters correctly, you need the experimentally determined voltage attenuation Ai,d (and Ad,i) for this, which is actually given by [(Vd-Vd,i)/Vd]. In case you decide to keep this paragraph, I would suggest to slightly change the indices in Eq.3. The use of gi and Ri is a little misleading as gi implies this is the membrane conductance at the position i. It would be clearer if you indicate already in the formula that gi is referring to the GABAeric conductance. And finally, the red stricken variables in the derivation of Eq. 3 look weird, like forgotten deleted items in the track-change mode of word. I wonder whether it is necessary to provide algebraic operations in this way in a manuscript.

6.) Impact of the results on active dendrites:

While I highly appreciate you approach to investigate the interaction between inhibitory synapses in a strictly dendro-centric approach, at the end I’m wondering how your findings will be affected by active properties of the dendrites. I would appreciate if you can include some exemplary experiments in the manuscript which demonstrate whether the inclusion of several active dendritic features will augment or ameliorate your findings. At least, you should discuss the impact of your findings for active dendrites.

Minor:

Line 19: While it became clear to me, what “accumulation […] of inhibitory effectiveness” means when reading the full text, this statement was really puzzling to me when first reading the abstract. Maybe it is possible to find a description for this observation that it easier to understand in the abstract.

Line 49: I would describe a dendrite, which is a physical entity of the neuron, more as a "neuronal part", "neuronal compartment" or "neuronal domain" rather than a "feature" of a neuron.

Line 75: In my humble opinion there are better references for the dominance of Cl- permeability in GABA(A) receptors available than these experiments in the crayfish muscle.

Line 85-102: Within this paragraph you used “synapses” and “inhibitory synapses” intermingled and synonymously. However, as in many studies inhibition was quantified by the influence of GABA on excitatory synapses, it was at first reading in some sentences ambiguous to me whether “synapse” may also refer to excitatory “test” synapses here. Therefore, I would suggest to use the phrase “inhibitory synapses” throughout this para.

Line 134 “50 ms otherwise”: In line 180 you mention that you determine IL for an integral from 0-150ms. Please specify.

Line 179 “Because the neuronal membrane is a capacitor and IL is calculated from Vm, the IL was taken at 150 ms unless otherwise stated”. To my understanding, this is not a valid explanation to choose exactly 150 ms as upper boarder for IL determination. Is there anything special with this time point regarding e.g. the time constant of the dendritic membrane or the stability of deltaEm or IL?

Line 218 and Fig. 2B, “NablaEGABA = -1 mV”: According to the definition in Formula #1 (line 122: nablaEGABA = VRest - EGABA) an nablaEGABA of -1mV means that EGABA was 1 mV larger (it is shifted more in the positive/depolarizing direction) than VRest.

Fig. 2D: In my pdf the colors of the 8 and 16 branch data look identical. Consider to use different colors, in this panel displaying only the shading information would also be sufficient.

Line 232 (Fig. 2D) “…AccIdx increased linearly…”: Are these indeed linear functions with different slopes for NablaEGABA of (-)1mV and (-)2 mV? Or are these slight deviations from a linear dependency that became apparent only for larger dendrite numbers?

Line 261-263, Fig. 3B: Here I like to know whether this interesting result is just caused by the mere amount of gGABA in adjacent position. In my opinion a more elusive approach would be to enhance the number of synapses in the “non-analyzed” branches and keep only 1 synapse in the analyzed branch. Does the results differ between one n x gGABA synapse in one adjacent dendrite and n gGABA synapses in n adjacent dendrites.

Line 265 “In addition, when some branches had more synapses than others (e.g. 150% effective number of synapses), the IL0 was naturally shared between them, but the ILd=i differed.”: In my opinion this sentence sounds strange, at it is obvious that ILd=i must differ (because gGABA differs between both sites). Consider rephrasing.

Line 270: Theoretically there should be no “saturation”, but an approximation towards hardly detectable voltage deflections if gGABA >> gInput. Consider rephrasing.

Line 283 “The branches without synapses ..”: Is this really "without synapses" or "without extra synapses"? From Fig. 3F-inset I assume that this should be “extra synapses”.

Line 314: Here you mention that more than 4 dendrites are required for an accumulation of IL at the junction, which is in contrast to Gidon & Segev (2012), who found that at least 3 inhibitory synapses are required for such an elevated inhibition. As long as you didn't test 3 branches, you should state here more carfully "but not in 2 branches".

Line 339, Fig. 4E: Here the color code suggested that the IL profile in the branches without synapses appeared to be identical in the "focal" and the "branch" situation. As this observation might have some functional impact, I wonder whether you already quantified whether such identical IL profiles indeed can be observed.

Line 356 “up until now”: Consider to be more specific here. E.g. "in Figs. 1-4" ?

Line 368: In you experiments it appears that EGABA approaches local values of -65 mV (Fig. 5D), suggesting that the transmembrane Cl- gradient breaks down. Does this happen under physiological conditions or is this just caused by the artificially long and strong GABAergic activity in your experimental paradigm. Please comment/discuss.

Line 401: Please consider rephrasing to emphasize that the maximum IL was obtained at X=0.5.

Line 446 “the greatest site of IL was…”. Consider rephrasing to "the site of greatest IL effect was …"

Line 450: Here you state that the “maximum level of IL” was reached at ca. 0.08X, whereas in line 415 an optimal IL distances of 0.07x was given (both for 4 dendrites and a EMFGABA of -5 mV). Please check and elaborate the description if there is an experimental difference between both simulations.

Line 480 “We find that GABAergic synapses with more negative EGABA are considerably better at suppressing local dendritic excitation (increased IL).” In my opinion this observation is probably not the main fining of your study (and also not a new and unexpected one). Therefore, this finding is probably not the best result to emphasize in the first sentences of the discussion. Consider to rephrase this section.

Line 507 “strongly affect IL and should be considered under a different framework that includes the time component of [Cl-]i loading (Eq. 1).” Please note that Eg. 1 did not explicitly include Cl- or IPSP.

Line 512: “…as the precise morphological structure (volume, surface area, compartmental ion differences, etc.)”. “Compartmental ion differences” are not really a morphological structure, but you may add other morphological features likes spines or tortuosity here.

Line 578 “… is no “V” marker….”: As you used the Nabla for identifying the EMF(GABA) I would suggest to use here the term "arrowhead" to avoid confusion.

Line 662: Please correct typo “placement”.

Fig. 7A, right panel. Here it is not clear to me how to interpret this graph. In the "branch" situation all synapses should distribute evenly along the dendrite, so which parameter is plotted here on the x axis and how can the complicated shape of the location dependency be explained?

Reference 31: Please correct “excitation in CA1 pyramidal neurons”.

Farrant & Kaila (2007) The cellular, molecular and ionic basis of GABA(A) receptor signaling. Prog Brain Res. 160:59-87

Gidon & Segev (2012) Principles governing the operation of synaptic inhibition in dendrites. Neuron 75:330-341

**Have the authors made all data and (if applicable) computational code underlying the findings in their manuscript fully available?**

Reviewer #1: Yes

Reviewer #2: Yes

PLOS authors have the option to publish the peer review history of their article (what does this mean?). If published, this will include your full peer review and any attached files.

Reviewer #1: **Yes: **Ryan S Phillips

Reviewer #2: No
---

## [Decision Letter · Decision Letter 1]

1 Jun 2022

Dear Dr Currin,

Thank you very much for submitting your manuscript "Computational models reveal how chloride dynamics determine the optimal distribution of inhibitory synapses to minimise dendritic excitability" for consideration at PLOS Computational Biology. As with all papers reviewed by the journal, your manuscript was reviewed by members of the editorial board and by several independent reviewers. The reviewers appreciated the attention to an important topic. Based on the reviews, we are likely to accept this manuscript for publication, providing that you modify the manuscript according to the review recommendations.

Sincerely,

Jonathan Rubin

Associate Editor

PLOS Computational Biology

Kim Blackwell

Deputy Editor

PLOS Computational Biology

[LINK]

Reviewer's Responses to Questions

**Comments to the Authors:**

Reviewer #1: Reviews have been uploaded as an attachment.

Reviewer #2: I appreciate that the authors considered all of my comments and provided more than sufficient responses.

Regarding the computation of Cl- and HCO3- dependency of reversal potentials and ion fluxes via GABA receptors, there are solutions and models in the NEURON environment available. However, I agree with the authors that their simplified solution is perfectly suited for their questions.

Line 20. Here the term "inhibitory accumulations" in citation marks reads strange. In my humble opinion it would be sufficient to state in the next sentence (line 21) "The extend of this inhibitory accumulation ...".

**Have the authors made all data and (if applicable) computational code underlying the findings in their manuscript fully available?**

Reviewer #1: Yes

Reviewer #2: Yes

PLOS authors have the option to publish the peer review history of their article (what does this mean?). If published, this will include your full peer review and any attached files.

Reviewer #1: No

Reviewer #2: No

Figure Files:

Data Requirements:

Reproducibility:

References:

---

## [Editor Report · Decision Letter 2]

1 Sep 2022

Dear Dr Currin,

We are pleased to inform you that your manuscript 'Computational models reveal how chloride dynamics determine the optimal distribution of inhibitory synapses to minimise dendritic excitability' has been provisionally accepted for publication in PLOS Computational Biology.

Best regards,

Jonathan Rubin

Academic Editor

PLOS Computational Biology

Kim Blackwell

Section Editor

PLOS Computational Biology

---

## [Editor Report · Acceptance letter]

16 Sep 2022

PCOMPBIOL-D-21-02224R2 

Computational models reveal how chloride dynamics determine the optimal distribution of inhibitory synapses to minimise dendritic excitability

Dear Dr Currin,

I am pleased to inform you that your manuscript has been formally accepted for publication in PLOS Computational Biology. Your manuscript is now with our production department and you will be notified of the publication date in due course.

With kind regards,

Zsanett Szabo
